# Sample Compression Scheme Reductions

**Idan Attias**                                                        IDANATTIAS88@GMAIL.COM
*University of Illinois at Chicago and Toyota Technological Institute at Chicago*

**Steve Hanneke**                                                      STEVE.HANNEKE@GMAIL.COM
*Purdue University*

**Arvind Ramaswami**                                                   RAMASWA4@PURDUE.EDU
*Purdue University*

**Editors:** Gautam Kamath and Po-Ling Loh

## Abstract

We present novel reductions from sample compression schemes in multiclass classification, regression, and adversarially robust learning settings to binary sample compression schemes. Assuming we have a compression scheme for binary classes of size $f(d_{\mathrm{VC}})$, where $d_{\mathrm{VC}}$ is the VC dimension, then we have the following results: (1) If the binary compression scheme is a majority vote or a stable compression scheme, then there exists a multiclass compression scheme of size $O(f(d_{\mathrm{G}}))$, where $d_{\mathrm{G}}$ is the graph dimension. Moreover, for general binary compression schemes, we obtain a compression of size $O(f(d_{\mathrm{G}}) \log |\mathcal{Y}|)$, where $\mathcal{Y}$ is the label space. (2) If the binary compression scheme is a majority vote or a stable compression scheme, then there exists an $\epsilon$-approximate compression scheme for regression over $[0,1]$-valued functions of size $O(f(d_{\mathrm{P}}))$, where $d_{\mathrm{P}}$ is the pseudo-dimension. For general binary compression schemes, we obtain a compression of size $O(f(d_{\mathrm{P}}) \log(1/\epsilon))$. These results would have significant implications if the sample compression conjecture, which posits that any binary concept class with a finite VC dimension admits a binary compression scheme of size $O(d_{\mathrm{VC}})$, is resolved (Littlestone and Warmuth, 1986; Floyd and Warmuth, 1995; Warmuth, 2003). Our results would then extend the proof of the conjecture immediately to other settings. We establish similar results for adversarially robust learning and also provide an example of a concept class that is robustly learnable but has no bounded-size compression scheme, demonstrating that learnability is not equivalent to having a compression scheme independent of the sample size, unlike in binary classification, where compression of size $2^{O(d_{\mathrm{VC}})}$ is attainable (Moran and Yehudayoff, 2016).

**Keywords:** Sample Compression Schemes, PAC Learning, Binary Classification, Multiclass Classification, Regression, Adversarially Robust Learning.

## 1. Introduction

A common guiding principle in machine learning is to favor simpler hypotheses when possible, following Occam's razor, which suggests that simpler models are more likely to generalize well. One approach to achieving simplicity, introduced by Littlestone and Warmuth (1986); Floyd and Warmuth (1995), is through a sample compression scheme for Probably Approximately Correct (PAC) learning (Valiant, 1984). This framework simplifies the process of hypothesis learning by compressing the information needed to represent a learned model. This is done by encoding the hypothesis using a small subset of the original training data (along with a short bit string), known as the *compression set*, and a *reconstruction* function that recovers from this subset a sample-consistent hypothesis on the entire training set. The size of the compression set reflects the complexity of the learning task, with smaller sets implying simpler models (in some sense). A well-known example

is the Support Vector Machine (SVM) algorithm, which constructs a halfspace in $\mathbb{R}^d$ using at most $d + 1$ support vectors to represent its decision boundary. A significant open problem in binary classification, known as the sample compression conjecture, proposes that any concept class with a finite VC dimension admits a compression scheme of size $O(d_{\mathrm{VC}})$, where $d_{\mathrm{VC}}$ is the VC dimension (Warmuth, 2003). A notable breakthrough by Moran and Yehudayoff (2016) demonstrated that every learnable binary concept class indeed admits a constant-size sample compression scheme (independent of the sample size), specifically of order $2^{O(d_{\mathrm{VC}})}$.

Beyond binary classification, sample compression schemes have also been explored in multiclass classification (Daniely et al., 2015; Daniely and Shalev-Shwartz, 2014; David et al., 2016; Brukhim et al., 2022; Pabbaraju, 2024). In particular, it is known that multiclass learnability with a finite set of labels is equivalent to having a constant-size compression scheme, with a compression size of $2^{O(d_{\mathrm{G}})}$ being achievable (David et al., 2016), where $d_{\mathrm{G}}$ denotes the graph dimension of the concept class. However, more recently, Pabbaraju (2024) demonstrated that this equivalence no longer holds when the label set is infinite, and in such cases, any sample compression must grow at least logarithmically with the sample size.

In the context of regression, there is a type of equivalence between learnability and sample compression (Hanneke et al., 2019; Attias et al., 2023, 2024). Specifically, in regression settings with respect to the $\ell_p$ loss, learnability has been shown to be equivalent to the existence of a bounded-size approximate compression scheme. In the realizable case, Hanneke et al. (2019) constructed an $\epsilon$-approximate compression scheme of size $\frac{1}{\epsilon} 2^{O(\mathrm{fat}_{c\epsilon})}$, for some constant $c > 0$, where $\mathrm{fat}_\gamma$ denotes the fat-shattering dimension (at scale $\gamma$) of the concept class. This can be extended to the agnostic setting, where the goal is for the approximate sample compression scheme to output a hypothesis with near-optimal error on the training data (with respect to the underlying concept class), rather than a sample-consistent hypothesis. Attias et al. (2024) showed that in this agnostic case, a compression scheme of size $\frac{1}{\epsilon} 2^{O(\mathrm{fat}_{c\epsilon})}$ can also be constructed. As with binary classification, determining the optimal size of a compression scheme for multiclass classification and regression remains a major open question.

In this paper, we explore reductions from sample compression schemes in multiclass classification, regression, and adversarially robust learning settings to binary sample compression schemes. Assuming the existence of a compression scheme for binary classes of size $f(d_{\mathrm{VC}})$, we construct compression schemes of approximately the same order for these more general settings, where $d_{\mathrm{VC}}$ is replaced by the appropriate dimension, in both realizable and agnostic settings. Our results would have significant implications if the sample compression conjecture were resolved, as this would allow us to extend the proof of the conjecture to other settings immediately. We summarize our contributions as follows.

## 1.1. Our Results

**Reductions from multiclass classification to binary classification (Section 3)**    Let $\mathcal{C} \subseteq \mathcal{Y}^{\mathcal{X}}$ be a multiclass concept class, with a finite graph dimension (denoted by $d_{\mathrm{G}}$, see Definition 3). Note that any multiclass concept class (with finite label space) is learnable if and only if its graph dimension is finite.

- We construct a sample compression scheme for multiclass classes of size $O(f(d_{\mathrm{G}}(\mathcal{C})) \log|\mathcal{Y}|)$ (Theorem 4).

- Assuming the reconstruction function of the compression scheme for binary classes either outputs a *majority vote* of concepts from $\mathcal{C}$ or selects a concept within the concept class $\mathcal{C}$ (*proper* compression), we construct a sample compression for $\mathcal{C}$ of size $O(f(d_{\mathrm{G}}(\mathcal{C})))$, even when infinite label sets are allowed (Theorem 7). This result is interesting in light of the fact that a primary method for constructing binary schemes uses majority votes.

- Assuming the existence of a *stable* sample compression scheme for binary concept classes, we construct a sample compression scheme for $\mathcal{C}$ of size $f(d_{\mathrm{G}}(\mathcal{C}))$ for finite label sets (Theorem 9). Stable compression ensures that removing any point outside the compression set does not affect the output of the compression function. A notable example of such a compression scheme is the SVM algorithm. For infinite label sets, we introduce an infinitized version of compression schemes, which allows us to prove analogous results (Theorem 11 and Theorem 13). Surprisingly, we show that a finite VC dimension is not sufficient for such a compression to exist, unlike standard compression schemes. Additionally, we show that while finite Littlestone dimension implies infinitized compression, the converse does not hold.

**Reductions from regression to binary classification (Section 4)** Let $\mathcal{C} \subseteq [0,1]^{\mathcal{X}}$ be a real-valued concept class, with a finite pseudo-dimension (denoted by $d_{\mathrm{P}}$, see Definition 15).

- We construct an $\epsilon$-approximate sample compression scheme for real-valued concept classes of size $O\big(f(d_{\mathrm{P}}(\mathcal{C})) \log \frac{1}{\epsilon}\big)$ for the $\ell_\infty$ loss, and $O\big(f(d_{\mathrm{P}}(\mathcal{C}))\frac{1}{p} \log \frac{1}{\epsilon}\big)$ for the $\ell_p$ loss, $p \in [1, \infty)$ (Theorem 16).

- Assuming binary concept classes have a compression scheme in one of the following forms: majority vote, proper, or stable compression scheme, we construct an $\epsilon$-approximate sample compression for real-valued concept classes of size $O(f(d_{\mathrm{P}}(\mathcal{C})))$ for any $\ell_p$ loss, $p \in [1, \infty]$ (Theorem 18 and Theorem 19).

- We demonstrate that, in certain cases, we can construct exact compression schemes for regression by using infinitized compression schemes (Theorem 21), or reduce the problem to multiclass classification with infinite labels (Theorem 22).

Note that our compression scales with the pseudo-dimension, which is known to be sufficient but not necessary for learnability. Whether we can have a similar reduction with the fat-shattering dimension is an open problem.

**Reductions from adversarially robust classification to binary classification (Section 5)** Let $\mathcal{C} \subseteq \{0,1\}^{\mathcal{X}}$ be a binary-valued concept class with a finite VC dimension. Let $\mathcal{U} : \mathcal{X} \to 2^{\mathcal{X}}$ be a perturbation function. In this setting, the loss of a concept $c$ on $(x, y)$ is $\sup_{z \in \mathcal{U}(x)} \mathbb{1}[c(z) \neq y]$.

- For bounded perturbation sets, letting $M = \sup_{x \in \mathcal{X}} |\mathcal{U}(x)|$, we construct an adversarially robust sample compression of size $O(f(d_{\mathrm{VC}}(\mathcal{C})) \log M)$ (Theorem 26). We get an improved compression size $O(f(d_{\mathrm{VC}}))$ if $\mathcal{C}$ admits a binary stable compression scheme (Theorem 27).

- We show that, in contrast to non-robust binary classification, where constant-size sample compression schemes exist for any learnable concept class, there is a robustly learnable concept class that does not admit any such scheme (Theorem 29). This negative result was observed also in multiclass classification (Pabbaraju, 2024) and list learning (Hanneke et al., 2024).

## 1.2. Other Related Work

Sample compression schemes have proven useful in a wide range of learning settings, particularly when the uniform convergence property either fails to hold or provides suboptimal rates. These applications include binary classification (Graepel et al., 2005; Moran and Yehudayoff, 2016; Bousquet et al., 2020), multiclass classification (Daniely et al., 2015; Daniely and Shalev-Shwartz, 2014; David et al., 2016; Brukhim et al., 2022), regression (Hanneke et al., 2018, 2019; Attias et al., 2023, 2024), active learning (Wiener et al., 2015), density estimation (Ashtiani et al., 2020), adversarially robust learning (Montasser et al., 2019, 2020, 2021, 2022; Attias et al., 2022a; Attias and Hanneke, 2023), learning with partial concepts (Alon et al., 2022), and demonstrating Bayes-consistency for nearest-neighbor methods (Gottlieb et al., 2014; Kontorovich et al., 2017). In fact, sublinear compressibility (with respect to sample size) and learnability are known to be equivalent for general learning problems (David et al., 2016).

A well-known approach for constructing sample compression schemes for general concept classes involves a weak-to-strong boosting procedure, where the resulting compression size is exponential in the combinatorial dimension of the problem in the worst case (Moran and Yehudayoff, 2016; David et al., 2016; Hanneke et al., 2019; Attias et al., 2024). These types of compressions are specifically referred to as majority vote compression schemes in this paper (see Definition 6). Another construction for general finite concept classes was provided by Moran et al. (2017). There is also extensive literature on improved compression schemes for specific cases, such as Floyd (1989); Helmbold et al. (1992); Floyd and Warmuth (1995); Ben-David and Litman (1998); Chernikov and Simon (2013); Kuzmin and Warmuth (2007); Rubinstein et al. (2009); Rubinstein and Rubinstein (2012); Livni and Simon (2013).

Bousquet et al. (2020) introduced the notion of stable compression schemes, whose choice of compression set is unaffected by removing points not in the compression set. The merit of such compression is that it provides an optimal generalization bound (improving by a log factor upon a generic compression scheme) for concept classes with such a scheme, for example, learning halfspaces with SVM, maximum classes, and intersection-closed classes. Hanneke and Kontorovich (2021) used similar techniques to provide novel or improved data-dependent generalization bounds for several learning problems.

## 2. Preliminaries

For any set $A$, define $A^*$ to be the set of finite sequences, where the elements are taken from $A$. For any $c : \mathcal{X} \to \mathcal{Y}$, a finite sequence $S = (x_1, y_1), \ldots, (x_n, y_n)$, and a loss function $\ell : \mathcal{Y} \times \mathcal{Y} \to [0, 1]$, define the empirical loss of $c$ on $S$ to be $L_S^\ell(c) = \frac{1}{n} \sum_{i=1}^n \ell(c(x_i), y_i)$. In multiclass and binary classification we use the zero-one loss $\ell_{0-1}(y, \widehat{y}) = \mathbb{1}[y \neq \widehat{y}]$. In regression we use the $\ell_p$ loss, $\ell_p(y, \widehat{y}) = |y - \widehat{y}|^p$, for $p \in [1, \infty)$. For $\ell_\infty$ loss, define $L_S^{\ell_\infty}(c) = \max_{1 \leq i \leq n} |c(x_i) - y_i|$. Additionally, $S$ is *realizable* if there exists $c \in \mathcal{C}$ such that $L_S^\ell(c) = 0$.

**Definition 1 (Sample Compression Schemes)** *Given a concept class $\mathcal{C} \subseteq \mathcal{Y}^\mathcal{X}$, define a sample compression scheme by the two following functions:*

- *A compression function $\kappa : (\mathcal{X} \times \mathcal{Y})^* \to (\mathcal{X} \times \mathcal{Y})^* \times \{0, 1\}^*$, which maps any underline{finite} sequence $S$ to a underline{finite} sequence (compression set) $S' \subseteq S$ and a underline{finite} bitstring $b$.[1]*

---

1. For anything of the form $S \subseteq T$, where either can be a sequence or a set, we will write $S \subseteq T$ to mean that $\{x : x \in S\} \subseteq \{x : x \in T\}$

- *A reconstruction function $\rho : (\mathcal{X} \times \mathcal{Y})^* \times \{0,1\}^* \to \mathcal{Y}^{\mathcal{X}}$ which maps any possible compression set to a predictor.*

  *The sample compression scheme is of size $k$ if for any sequence $S$, for $\kappa(S) = (S', b)$, it holds that $|S'| + |b| \leq k$.*

*Given a loss function $\ell : \mathcal{Y} \times \mathcal{Y} \to [0,1]$, a concept class $\mathcal{C} \subseteq \mathcal{Y}^{\mathcal{X}}$, and $\epsilon > 0$, consider the following types of sample compression schemes:*

- *Exact Agnostic:* *For all finite $S$, $L_S^\ell(\rho(\kappa(S))) \leq \inf_{c \in \mathcal{C}} L_S^\ell(c)$.*

- *Exact Realizable:* *For all finite realizable $S$, $L_S^\ell(\rho(\kappa(S))) = 0$.*

- *$\epsilon$-approximate Agnostic:* *For all finite $S$, $L_S^\ell(\rho(\kappa(S))) \leq \inf_{c \in \mathcal{C}} L_S^\ell(c) + \epsilon$.*

- *$\epsilon$-approximate Realizable:* *For all finite realizable $S$, $L_S^\ell(\rho(\kappa(S))) \leq \epsilon$.*

Unless explicitly specified in this paper, when referring to "sample compression schemes," we mean exact realizable sample compression schemes using the zero-one loss function, i.e., for a realizable sequence $S = (x_1, y_1), (x_2, y_2), \ldots, (x_n, y_n)$, $\rho(\kappa(S))$ outputs a sample-consistent predictor: $\rho(\kappa(S))(x_i) = y_i$ for all $1 \leq i \leq n$. Additionally, if it is not clarified whether the compression scheme is exact or approximate, the compression scheme can be assumed to be exact.

**Definition 2 (VC Dimension (Vapnik and Chervonenkis, 1971))** *We say that $x_1, \ldots, x_n \in \mathcal{X}$ are shattered by $\mathcal{C} \subseteq \{0,1\}^{\mathcal{X}}$ if $\{(c(x_1), \ldots, c(x_n)) : c \in \mathcal{C}\} = \{0,1\}^n$. The Vapnik-Chervonenkis (VC) dimension of a binary concept class $\mathcal{C}$, denoted by $d_{\mathrm{VC}}(\mathcal{C})$, is the largest nonnegative integer $n \in \mathbb{N}$ for which there exist $x_1, \ldots, x_n \in \mathcal{X}$ that are shattered in $\mathcal{C}$.*

The sample compression conjecture (Littlestone and Warmuth, 1986; Floyd and Warmuth, 1995; Warmuth, 2003) states that for classes with finite VC dimension, there exists a compression scheme of size $O(d_{\mathrm{VC}})$.

## 3. Compression for Multiclass Classification

In this section, we tackle the problem of multiclass compression by reducing it to the binary setting. In multiclass classification, the finiteness of the graph dimension of a concept class characterizes learnability when the label set is finite (albeit with non-optimal sample complexity in general). David et al. (2016) demonstrated that, in such cases, a sample compression scheme of size $2^{O(d_{\mathrm{G}})}$ can be constructed, with the compression size notably independent of the sample size. However, Pabbaraju (2024) showed that when the label set is infinite, there exist concept classes where any sample compression scheme must grow at least logarithmically with the sample size.

Therefore, we explore reductions from multiclass compression to binary compression under the assumption of a finite graph dimension. It is important to note that the graph dimension alone is sufficient but not necessary for multiclass learnability when the number of labels is infinite.

**Definition 3 (Graph Dimension (Natarajan, 1989; Ben-David et al., 1992))** *A set of points $x_1, \ldots, x_n \in \mathcal{X}$ is G-shattered by $\mathcal{C} \subseteq \mathcal{Y}^{\mathcal{X}}$ if there exist $y_1, \ldots, y_n \in \mathcal{Y}$ such that*

$$\{(\mathbb{1}[c(x_1) = y_1], \mathbb{1}[c(x_2) = y_2], \ldots, \mathbb{1}[c(x_n) = y_n]) : c \in \mathcal{C}\} = \{0,1\}^n.$$

*The graph dimension of a multiclass concept class $\mathcal{C}$, denoted by $d_\mathrm{G}(\mathcal{C})$, is the largest nonnegative integer $n \in \mathbb{N}$ for which there exist $x_1, \ldots, x_n \in \mathcal{X}$ that are G-shattered by $\mathcal{C}$.*

Consider a multiclass concept class $\mathcal{C} \subseteq \mathcal{Y}^{\mathcal{X}}$. For any $S = (x_1, y_1), (x_2, y_2), \ldots (x_n, y_n)$ realizable by $\mathcal{C}$, define the "inflated" set $S_{\mathcal{Y}}$ as follows

$$S_{\mathcal{Y}} = \{((x_i, y), \mathbb{1}[y = y_i]) : i \in [n], y \in \mathcal{Y}\}. \tag{1}$$

Define $\mathcal{C}_{\mathcal{Y}}$ as follows,

$$\mathcal{C}_{\mathcal{Y}} = \{g_c : c \in \mathcal{C}\}, \tag{2}$$

where $g_c : \mathcal{X} \times \mathcal{Y} \to \{0, 1\}$ is defined such that $g_c(x, y) = \mathbb{1}[c(x) = y]$. This inflation operation transforms the multiclass prediction problem into a binary classification problem: for each original example $(x_i, y_i)$, we create $|\mathcal{Y}|$ binary-labeled examples, where the positive label indicates the correct class. Similarly, each multiclass concept $c$ is transformed into a binary concept $g_c$ that identifies correct label predictions. In the following, we construct a multiclass sample compression scheme for classes of finite labels and finite graph dimension, via a reduction to the binary setting.

**Theorem 4 (Reducing Multiclass Compression Schemes to Binary Compression Schemes)**
*Suppose that for binary concept classes with finite VC dimension $d_\mathrm{VC} < \infty$, there exists a sample compression scheme of size $f(d_\mathrm{VC})$. Then, for multiclass concept classes with a finite label set $|\mathcal{Y}|$ and a graph dimension $d_\mathrm{G} < \infty$, there exists a sample compression scheme of size $O(f(d_\mathrm{G}) \log |\mathcal{Y}|)$.*

**Proof** Let $\mathcal{C} \subseteq \mathcal{Y}^{\mathcal{X}}$ be a multiclass concept class. Consider the inflated dataset $S_{\mathcal{Y}}$ (Equation 1) and the class $\mathcal{C}_{\mathcal{Y}}$ (Equation 2). Denote $d_\mathrm{VC}(\mathcal{C}_{\mathcal{Y}}) = d_\mathrm{VC}$ and $d_\mathrm{G}(\mathcal{C}) = d_\mathrm{G}$. It is straightforward to show that $d_\mathrm{VC} = d_\mathrm{G}$. To show the $\leq$ direction, we have that for any $(x_1, y_1), \ldots, (x_n, y_n)$ shattered by $\mathcal{C}_{\mathcal{Y}}$, all the $x_i$'s must be distinct (otherwise, there is an $(x, y_{j_1}), (x, y_{j_2})$ with $y_{j_1} \neq y_{j_2}$ that are shattered, and are both assigned value 1 by a concept in $\mathcal{C}_{\mathcal{Y}}$, which implies that there exists an $c \in \mathcal{C}$ such that $c(x) = y_{j_1}$ and $c(x) = y_{j_2}$, which is not possible). Thus, $x_1, \ldots, x_n$ are G-shattered in $\mathcal{C}$ via labels $y_1, \ldots, y_n$. To show the $\geq$ direction, consider $x_1, \ldots, x_n$ G-shattered in $\mathcal{C}$ via labels $y_1, \ldots, y_n$. It is clear that $(x_1, y_1), \ldots, (x_n, y_n)$ are shattered in $\mathcal{C}_{\mathcal{Y}}$.

Suppose $\mathcal{C}_{\mathcal{Y}}$ has a binary compression scheme $(\kappa_b, \rho_b)$, then we construct a compression scheme $(\kappa, \rho)$ for $\mathcal{C}$ as follows. **Compression:** Given a dataset $S = \{(x_1, y_1), \ldots, (x_n, y_n)\}$ realizable by $\mathcal{C}$, construct $\kappa(S)$ as follows. Inflate $S$ to $S_{\mathcal{Y}}$. Note that $S_{\mathcal{Y}}$ is realizable by $\mathcal{C}_{\mathcal{Y}}$, and $d_\mathrm{VC} = d_\mathrm{G}$. We can apply $\kappa_b$ to $S_{\mathcal{Y}}$ to get a compression of size $f(d_\mathrm{G})$. The compression points will be of the form $((x, y), z), z \in \{0, 1\}$. For the points where $z = 1$, we have that $(x, y) = (x_i, y_i)$ for some $i$, so we can add that to $\kappa(S)$, contributing 1 for each. For the points where $z = 0$, we need $\log |\mathcal{Y}|$ bits to add that to $\kappa(S)$. Thus, our compression size will be $\leq f(d_\mathrm{G}) + f(d_\mathrm{G}) \log |\mathcal{Y}| = O(f(d_\mathrm{G}) \log |\mathcal{Y}|)$. **Reconstruction:** Our compression has enough information for us to retrieve the result of $\kappa_b(\mathcal{C}_{\mathcal{Y}})$. We can directly apply $\rho_b$ on this to get the desired result. ∎

For classes with graph dimension 1 we can get a tighter result, but for general concept classes and binary compression schemes it is an open problem whether we can remove the $\log|\mathcal{Y}|$ factor from the compression size in Theorem 4.

**A sample compression scheme for graph dimension** 1    We show that any concept class $\mathcal{C}$ with graph dimension 1 admits a sample compression scheme of size 1. The proof is in Appendix B. The idea is based on a technique of Ben-David (2015) which established a sample compression scheme of size 1 for binary classes with VC dimension 1. Although our result has been previously shown by Samei et al. (2014), the proof presented here uses a different technique and may offer additional insights.

**Open Problem 5** *Suppose all binary concept classes with VC dimension $d_{\mathrm{VC}}$ have a sample compression scheme of size $f(d_{\mathrm{VC}})$. Does every multiclass class with graph dimension $d_{\mathrm{G}}$ have a sample compression scheme of size $O(f(d_{\mathrm{G}}))$?*

### 3.1. Additional Assumption: Existence of Proper or Majority Vote Binary Compression

We can derive tighter results for sample compression schemes with particular reconstruction functions, such as majority votes of concepts in the class. Majority votes are a natural choice for reconstruction functions, as many known sample compression schemes are based on boosting methods with such a property (Moran and Yehudayoff, 2016; David et al., 2016; Hanneke et al., 2019; Attias et al., 2024). We also define a proper compression scheme where the reconstruction function returns a concept from the class.

**Definition 6** *Given a concept class $\mathcal{C}$, a compression scheme is a proper compression scheme if for every finite $S \in (\mathcal{X} \times \mathcal{Y})^*$, the reconstruction $\rho(\kappa(S))$ returns a concept $c \in \mathcal{C}$.*

*A binary function $f : \mathcal{X} \to \{0,1\}$ is a majority of concepts from $\mathcal{C} \subseteq \{0,1\}^{\mathcal{X}}$ if there exists a finite $\mathcal{C}_f \subseteq \mathcal{C}$ such that for all $x \in \mathcal{X}$, $f(x) = \mathrm{Maj}(c(x) : c \in \mathcal{C}_f)$, where $\mathrm{Maj}(\cdot)$ takes in a sequence and returns the majority element (picking zero to break ties). A compression scheme is a majority vote compression scheme if for every finite $S \in (\mathcal{X} \times \mathcal{Y})^*$, $\rho(\kappa(S))$ outputs a majority of concepts from $\mathcal{C}$.*

### Theorem 7 (Multiclass, Reductions with Proper / Majority Vote Compression Schemes)
*Suppose any binary concept class $\mathcal{C}$ with VC dimension $d_{\mathrm{VC}} < \infty$ has a compression scheme of size $f(d_{\mathrm{VC}})$ that is either a proper compression scheme or a majority vote compression scheme (see Definition 6). Then any multiclass concept class (allowing infinite label sets) with a finite graph dimension $d_{\mathrm{G}} < \infty$ admits a compression scheme of size $O(f(d_{\mathrm{G}}))$.*

In particular, we recover the best known bounded-size multiclass compression scheme of size $2^{O(d_{\mathrm{G}})}$ (David et al., 2016), via a reduction to binary compression.

**Proof** Suppose $\mathcal{C}_{\mathcal{Y}}$ (Equation 2) has a binary compression scheme $(\kappa_b, \rho_b)$, which is either a proper compression scheme or a majority vote compression scheme. Consider $h = \rho_b(T)$ for any $T$ in the image of $\kappa_b$ and any possible finite realizable $S$ given to $\kappa_b$ as an input. We claim that for all $x \in \mathcal{X}$, $h(x, y)$ equals 1 for at most one $y$. To show this, we consider two cases: 1) $h$ is proper, and 2) $h$ is a majority of concepts from the concept class. In the first case, since $h$ is proper in $\mathcal{C}_{\mathcal{Y}}$, $h(x, y)$ must equal 1 for exactly one $y \in \mathcal{Y}$. Now consider the second case, where $h$ is a majority of concepts $\mathcal{C}_h \subseteq \mathcal{C}$. Notice that for any fixed $x \in \mathcal{X}$, $c(x, y) = 1$ for exactly $|\mathcal{C}_h|$ pairs $(c, y)$ in $\mathcal{C}_h \times \mathcal{Y}$ (since for any $c \in \mathcal{C}_h$, there exists exactly one $y \in \mathcal{Y}$ such that $c(x, y) = 1$). Thus, in order for $h(x, y)$ to be 1, $c(x, y)$ must be 1 for strictly more than $|\mathcal{C}_h|/2$ concepts $c$ from $\mathcal{C}_h$. Therefore, $h(x, y) = 1$ for at most one $y \in \mathcal{Y}$.

We now construct a compression scheme $(\kappa, \rho)$ for $\mathcal{C}$ as follows. **Compression:** Given realizable $S = (x_1, y_1), \ldots, (x_n, y_n)$, let $T = \{((x_i, y_i), 1) : i \in [n]\}$. We let our compression $\kappa(S)$ return the $(x, y)$ pairs from $\kappa_b(T)$, which have size $O(f(d_{\mathrm{G}}))$. **Reconstruction:** We have that $\kappa_b(T) = \{((x, y), 1) : (x, y) \in \kappa(S)\}$, so we can immediately recover $\kappa_b(T)$. Since $\rho_b(\kappa_b(T))$ is correct on all possible $T$ and predicts 1 for at most one $y$ for each $x$, we can conclude that $\rho_b(\kappa_b(T))$ is correct on all of $S_{\mathcal{Y}}$. Note that if the reconstruction $\rho_b$ is proper, then it will output exactly one 1 for each $x$. Additionally, it the reconstruction is a majority of learners from the class, where we break ties to favor predicting 0, then it will output at most one 1 for each $x$. ∎

### 3.2. Additional Assumption: Existence of Stable Binary Compression

By assuming the existence of stable sample compression schemes for binary classification, we can derive tighter results, including a reduction that is independent of the size of the label space, assuming the label space is finite. A stable compression scheme ensures that removing any point outside the compression set does not affect the output of the compression function. Natural examples where such schemes exist include thresholds, halfspaces, maximum classes, and intersection-closed classes. A notable example for halfspaces is the Support Vector Machine (SVM) algorithm: points outside the $d + 1$ support vectors can be removed, and the remaining support vectors still form a valid compression set. The formal definition is as follows.

**Definition 8 (Stable Compression Schemes (Bousquet et al., 2020))** *A sample compression scheme is stable if for any sequence $S$ over $(\mathcal{X} \times \mathcal{Y})$, and any $T : \kappa(S) \subseteq T \subseteq S$, it holds that $\kappa(T) = \kappa(S)$ [2].*

**Theorem 9 (Multiclass, Reductions with Stable Compression Schemes)** *Suppose that for binary concept classes with finite VC dimension $d_{\mathrm{VC}} < \infty$, there exists a stable sample compression scheme of size $f(d_{\mathrm{VC}})$. Then, for multiclass concept classes with a finite graph dimension $d_{\mathrm{G}} < \infty$ and finite label space, there exists a stable sample compression scheme of size $O(f(d_{\mathrm{G}}))$.*

**Proof** Given a class $\mathcal{C}$ with $d_G(\mathcal{C}) = d_G$, consider $\mathcal{C}_{\mathcal{Y}}$ (Equation 2). We construct the following compression scheme. **Compression:** We first inflate $S$ to $S_{\mathcal{Y}}$ and apply $\kappa_b$. Define

$$\kappa_b'(S_{\mathcal{Y}}) := \{((x, y), \mathbb{1}[y = y_i]) : x \text{ appears in } \kappa_b(S_{\mathcal{Y}}), (x, y_i) \in S, y \in \mathcal{Y}\},$$

i.e., it is an inflated $\kappa_b(S_{\mathcal{Y}})$ to include all the labels for each $x$. Since the compression scheme is stable, and since $\kappa_b(S_{\mathcal{Y}}) \subseteq \kappa_b'(S_{\mathcal{Y}}) \subseteq S_{\mathcal{Y}}$, we must have that $\kappa_b(\kappa_b'(S_{\mathcal{Y}})) = \kappa_b(S_{\mathcal{Y}})$. We can let $\kappa(S)$ return the $(x_i, y_i)$ pairs such that $x_i$ is a member of $\kappa_b(S_{\mathcal{Y}})$. This will have size $O(f(d_{\mathrm{G}}))$. **Reconstruction:** Given $\kappa(S)$, we can reconstruct $\kappa_b'(S_{\mathcal{Y}})$ by inflating the dataset to include all labels for each $x$ in $\kappa(S)$. Then, we can apply $\kappa_b$ to get $\kappa_b(S_{\mathcal{Y}})$ (using the fact that it is a stable compression scheme), and then apply $\rho_b$ to get the desired compression result.

---

2. Some notations:

- If there is a $\subseteq$ sign and any of the sides is of the form $(S, b)$ for a sequence $S$ and a bitstring $b$, we can interpret it to be $S$. For example, $(S, b) \subseteq T$ will be interpreted as $S \subseteq T$.

- Also, we will sometimes apply $\kappa$ to something of the form $(T, b)$ (for example, something like $\kappa(\kappa(S))$). In this case, we can interpret it to be $\kappa(T)$.

We now show that the above compression scheme is stable. Let $(\kappa, \rho)$ be the multiclass compression scheme above, and let $(\kappa_b, \rho_b)$ be the binary stable compression scheme above. Let $S = (x_1, y_1), \ldots, (x_n, y_n)$ be realizable, and consider $T$ such that $\kappa(S) \subseteq T \subseteq S$. We want to show that $\kappa(T) = \kappa(S)$. Inflate $S$ to $S_\mathcal{Y}$ as above, and inflate $T$ to $T_\mathcal{Y}$ similarly. $\kappa(S) \subseteq T$, and in the inflated dataset $T_\mathcal{Y}$ includes all the labels for each $x \in T$, so $\kappa_b(S_\mathcal{Y}) \subseteq T_\mathcal{Y}$. Additionally since $T \subseteq S$, we get

$$\kappa_b(S_\mathcal{Y}) \subseteq T_\mathcal{Y} \subseteq S_\mathcal{Y}.$$

Since the binary compression scheme is stable, this gives that $\kappa_b(T_\mathcal{Y}) = \kappa_b(S_\mathcal{Y})$. Thus, $\kappa(T) = \kappa(S)$, as desired. ∎

We prove a similar result for infinite label sets, where we require that the compression scheme on $\mathcal{C}_\mathcal{Y}$ (Equation 2) can handle infinite sets. For a set $A$, we denote by $A^\infty$ the set of (possibly infinite) sequences whose elements are taken from $A$. A (possibly infinite) sequence $S \in (\mathcal{X} \times \mathcal{Y})^\infty$ is *realizable* if there exists a $c \in \mathcal{C}$ such that for all $(x, y) \in S$, $c(x) = y$. We define infinite compression schemes as follows:

**Definition 10 (Infinitized Sample Compression Scheme)**  *Given a concept class $\mathcal{C} \subseteq \mathcal{Y}^\mathcal{X}$, define an infinitized sample compression scheme by the two following functions:*

- *A compression function $\kappa : (\mathcal{X} \times \mathcal{Y})^\infty \to (\mathcal{X} \times \mathcal{Y})^* \times \{0, 1\}^*$ which maps any (possibly infinite) sequence $S$ to a finite sequence (compression set) $S' \subseteq S$ and a finite bitstring $b$.*

- *A reconstruction function $\rho : (\mathcal{X} \times \mathcal{Y})^* \times \{0, 1\}^* \to \mathcal{Y}^\mathcal{X}$, which maps any possible compression set to a predictor.*

*Additionally, for any realizable $S \in (\mathcal{X} \times \mathcal{Y})^\infty$, $\rho(\kappa(S))(x) = y$ for all $(x, y) \in S$. The infinitized sample compression scheme is of size $k$ if for any realizable sequence $S \in (\mathcal{X} \times \mathcal{Y})^\infty$, for $\kappa(S) = (S', b)$, $|S'| + |b| \leq k$.*

Appendix A includes a discussion of standard and infinitized sample compression schemes, where we demonstrate the distinctions between these two notions. We show that a finite VC dimension is not sufficient for the existence of a bounded-size infinitized sample compression scheme. For example, thresholds on the real line have a finite VC dimension but do not admit an infinitized compression scheme. Additionally, we show that a finite Littlestone dimension is sufficient (but not necessary) for a bounded-size infinitized compression scheme. We leave as an open problem the question of characterizing which concept classes admit a bounded-size infinitized compression scheme.

One can study infinitized compression schemes in more general settings, such as compression with general losses, agnostic compression, and approximate compression. We do not attempt to do so in this paper, and we focus instead on exact realizable infinitized compression with the zero-one loss.

**Theorem 11 (Multiclass, $|\mathcal{Y}| = \infty$, Reductions with Stable Infinitized Compression Schemes)**  *Let $\mathcal{C} \subset \mathcal{Y}^\mathcal{X}$ be a multiclass concept class. If $\mathcal{C}_\mathcal{Y}$ (see Equation 2) has an infinitized stable compression scheme of size $k$, then $\mathcal{C}$ has a stable compression scheme of size $k$.*

The idea to prove the theorem is to consider an infinitized compression scheme $(\kappa_b, \rho_b)$ over the infinite set $S_{\mathcal{Y}}$. The result follows directly via the arguments of the proof of Theorem 9, the details are omitted here to avoid repetition.

The requirement for a sample compression scheme to be infinitized is rather strong since we require $\rho(\kappa(S))$ to be consistent with any *finite or infinite* $S$. Instead, we relax the definition, by defining an "inflated" compression scheme, which will be defined below formally. The idea is that instead of considering all infinite sets $S$, we only require the accuracy guarantee to hold for sets $S$ that are supported by a finite number of $x \in \mathcal{X}$.

**Definition 12 (Inflated Compression Scheme)** *Given a concept class $\mathcal{C} \subset \{0,1\}^{(\mathcal{X} \times \mathcal{Y})}$, define an inflated compression scheme by the following two functions:*

- *A compression function $\kappa : (\mathcal{X} \times \mathcal{Y} \times \{0,1\})^{\infty} \to (\mathcal{X} \times \mathcal{Y} \times \{0,1\})^* \times \{0,1\}^*$ which maps any (possibly infinite) sequence $S$ to a finite sequence (compression set) $S' \subseteq S$ and a finite bitstring $b$.*

- *A reconstruction function $\rho : (\mathcal{X} \times \mathcal{Y} \times \{0,1\})^* \times \{0,1\}^* \to \{0,1\}^{(\mathcal{X} \times \mathcal{Y})}$, which maps any possible compression set to a predictor.*

*Additionally, for any realizable sequence $S \in (\mathcal{X} \times \mathcal{Y} \times \{0,1\})^{\infty}$ where $|\{x \in \mathcal{X} : \exists y \in \mathcal{Y}, z \in \{0,1\} \text{ s.t. } ((x,y),z) \in S\}| < \infty$, it must hold that $\rho(\kappa(S))((x,y)) = z$ for all $((x,y),z) \in S$. The inflated sample compression scheme is of size $k$ if for any such realizable sequence $S$, where $\kappa(S) = (S', b)$, we have $|S'| + |b| \le k$.*

**Theorem 13 (Multiclass, $|\mathcal{Y}| = \infty$, Reductions with Stable Inflated Compression Schemes)**
*Let $\mathcal{C} \subset \mathcal{Y}^{\mathcal{X}}$ be a multiclass concept class. If $\mathcal{C}_{\mathcal{Y}}$ (see Equation 2) has a stable inflated compression scheme of size $k$, then $\mathcal{C}$ has a stable compression scheme of size $k$.*

The proof follows similar reasoning as Theorem 11 and is omitted here for brevity. Inflated compression schemes are more practical, and there are natural settings where such compression schemes are relevant. For example, consider the following example.

**Example 1 (Concept Class with Inflated Stable Compression Scheme: Piecewise Thresholds)**
*Consider the class $\mathcal{C} \subseteq \mathcal{Y}^{\mathcal{X}}$, where $\mathcal{X} = \mathcal{Y} = \mathbb{R}$, of two piecewise thresholds, i.e. $\{g_{t,y_1,y_2} : t, y_1, y_2 \in \mathbb{R}\}$ where*

$$g_{t,y_1,y_2}(x) = \begin{cases} y_1 & \text{if } x \le t, \\ y_2 & \text{if } x > t. \end{cases}$$

*For a concept $c$, the class $\mathcal{C}_{\mathcal{Y}}$ will map $(x,y)$ to 1 or 0 depending on whether $c(x) = y$. We construct a stable inflated compression scheme for this class. For a set $S$ of points of the form $((x,y),z)$, discard the points for which there is no other point with the corresponding $x$ value and the label equal to 1. Then, for each $y \in \mathcal{Y}$ that appears in the set, compress to the leftmost and rightmost datapoints of the form $((x,y),1)$. To predict the value at an $(x,y)$ pair, we can check the compression scheme to find the leftmost and rightmost points $x_l, x_r$ in $\mathcal{X}$ for which $y$ occurs. If $x_l \le x \le x_r$, predict 1, and otherwise, predict 0. This gives a compression scheme of size 4, that is also stable, since removing points from $S$ that are not in $\kappa(S)$ will not affect the output of $\kappa$. This proof can be generalized to the class of $k$ piecewise thresholds, which has an inflated stable compression scheme of size $2k$.*

### 3.3. Agnostic Multiclass Sample Compression Scheme

We show that our results simply extend to agnostic sample compression (see Definition 1).

**Theorem 14 (Reducing Agnostic Multiclass Compression to Binary Compression Schemes)**
*Suppose that for binary classes with VC dimension $d_{\mathrm{VC}} < \infty$, there exists a sample compression scheme of size $f(d_{\mathrm{VC}})$. Then, for a multiclass class with a finite graph dimension $d_{\mathrm{G}} < \infty$ and a finite label set $|\mathcal{Y}|$, there exists an agnostic sample compression scheme of size $O(f(d_{\mathrm{G}}) \log |\mathcal{Y}|)$.*

**Proof** Consider $S = (x_1, y_1), \ldots, (x_n, y_n)$. Run Empirical Risk Minimization (ERM) to compute an $\widehat{c}$ that minimizes the empirical loss:

$$\inf_{c \in \mathcal{C}} \frac{1}{n} \sum_{i=1}^{n} \mathbb{1}[c(x_i) \neq y_i].$$

Denote the examples on which $\widehat{c}$ is correct as $S' = (x_{i_1}, y_{i_1}), \ldots, (x_{i_k}, y_{i_k})$, for some indices $i_1, i_2, \ldots, i_k$. Consider $\mathcal{C}_{\mathcal{Y}}$ (see Equation 2). We can inflate the labels to $S'_{\mathcal{Y}} = \{((x_{i_j}, y), \mathbb{1}[y_{i_j} = y]) : 1 \leq j \leq k\}$. This is realizable by $\mathcal{C}_{\mathcal{Y}}$, so we can apply the binary compression scheme to this class and construct our multiclass compression scheme as in the proof of Theorem 4. ∎

We can prove the agnostic version of Theorem 7 and Theorem 9 similarly, by finding the largest realizable subsequence in the training set and applying it for the realizable compression scheme on this subsequence.

## 4. Compression for Regression

In this section, we tackle the problem of compression in a regression setting with the $\ell_p$ loss by reducing it to the binary setting. Our compression size depends on the pseudo-dimension, which is known to be sufficient but not necessary for learnability. We leave as an important open problem the question of whether our reductions could work for concept classes with a finite fat-shattering dimension. It is known that (without reductions) it is possible to construct such compression schemes of size $\frac{1}{\epsilon} 2^{O(\mathrm{fat}_{c\epsilon})}$ (for some $c > 0$), in both realizable and agnostic settings (Hanneke et al., 2019; Attias et al., 2024). The pseudo-dimension is defined as follows.

**Definition 15 (Pseudo-Dimension (Pollard, 1984, 1990))** *A set of points $x_1, x_2, \ldots, x_n$ is P-shattered by $\mathcal{C} \subseteq [0, 1]^{\mathcal{X}}$ if there exist $y_1, y_2, \ldots, y_n \in [0, 1]$ such that*

$$\{(\mathbb{1}[c(x_1) \leq y_1], \mathbb{1}[c(x_2) \leq y_2], \ldots, \mathbb{1}[c(x_n) \leq y_n]) : c \in \mathcal{C}\} = \{0, 1\}^n.$$

*The pseudo-dimension of a real-valued concept class $\mathcal{C}$, denoted by $d_{\mathrm{P}}(\mathcal{C})$ is the largest nonnegative integer $n \in \mathbb{N}$ for which there exist $x_1, x_2, \ldots, x_n \in \mathcal{X}$ that are P-shattered by $\mathcal{C}$.*

Let $\mathcal{C} \subseteq [0, 1]^{\mathcal{X}}$ be a real-valued class. Let $\mathcal{C}_{\leq}$ consist of functions $g_c : \mathcal{X} \times [0, 1] \to \{0, 1\}$ where $g_c(x, y) = \mathbb{1}[c(x) \leq y]$. For any $\epsilon \in (0, 1)$, define the $\epsilon$-discretized label set $\mathcal{Y}_\epsilon$ to be

$$\mathcal{Y}_\epsilon = \left\{ c\epsilon : c \in \left\{ 0, 1, 2, \ldots, \left\lfloor \frac{1}{\epsilon} \right\rfloor \right\} \right\} \cup \{1\}. \tag{3}$$

For any realizable sequence $S = (x_1, y_1), (x_2, y_2), \ldots, (x_n, y_n)$ in $(\mathcal{X} \times [0, 1])^n$, and for any $\epsilon \in (0, 1)$, let $\mathcal{S}_\epsilon$ be defined as the following inflated dataset:

$$S_\epsilon = \bigcup_{i=1}^{n} \{((x_i, y), \mathbb{1}[y_i \leq y]) : y \in \mathcal{Y}_\epsilon\}, \tag{4}$$

corresponding to the output of a sample-consistent concept from $\mathcal{C}_\leq$ on $\mathcal{X} \times \mathcal{Y}_\epsilon$.

**Theorem 16 (Reducing (Approximate) Compression for $\ell_p$ Regression to Binary)** *Suppose that for binary classes with VC dimension $d_{\mathrm{VC}} < \infty$, there exists a sample compression scheme of size $f(d_{\mathrm{VC}})$. Then, for a $[0, 1]$-real-valued class with pseudo-dimension $d_{\mathrm{P}} < \infty$, there is an $\epsilon$-approximate compression scheme with respect to the $\ell_\infty$ loss with size $O(f(d_{\mathrm{P}}) \log(\frac{1}{\epsilon}))$. Furthermore, for $p \in [1, \infty)$, there is an $\epsilon$-approximate compression scheme with respect to the $\ell_p$ loss of size $O\left(\frac{f(d_{\mathrm{P}}) \log(\frac{1}{\epsilon})}{p}\right)$.*

**Proof** Denote the pseudo-dimension $d_{\mathrm{P}}(\mathcal{C}) = d_{\mathrm{P}}$. We first consider a compression scheme for $\ell_\infty$ loss. We show that $d_{\mathrm{VC}}(\mathcal{C}_\leq) = d_{\mathrm{P}}$. First, we show the $\leq$ direction. Consider $(x_1, y_1), (x_2, y_2), \ldots, (x_n, y_n)$ shattered by the binary class $\mathcal{C}_\leq$. All the $x_i$'s must be distinct, as otherwise, two points $(x, y_{j_1}), (x, y_{j_2})$ with $y_{j_1} < y_{j_2}$ would require $g_c(x, y_{j_2}) < g_c(x, y_{j_1})$ for some $g_c$ to achieve shattering, contradicting the monotonicity of $g_c$ in $y$ for fixed $x \in \mathcal{X}$. Since the $n$ points are shattered, this means that for all $b \in \{0, 1\}^n$, there exists a $g_c \in \mathcal{C}_\leq$ such that $g_c(x_i, y_i) = b_i$. By the definition of $g_c$, this implies that for all $b \in \{0, 1\}^n$, there exists a $c \in \mathcal{C}$ such that for all $i = 1, \ldots, n$, if $b_i = 1$, then $c(x_i) \leq y_i$, and if $b_i = 0$, then $c(x_i) > y_i$. Thus, $x_1, x_2, \ldots, x_n$ P-shatter $\mathcal{C}$ via labels $y_1, y_2, \ldots, y_n$. Now we show the $\geq$ direction. Suppose $x_1, x_2, \ldots, x_n$ P-shatter $\mathcal{C}$ via labels $y_1, y_2, \ldots, y_n$. This implies that for all $b \in \{0, 1\}^n$, there exists a $c \in \mathcal{C}$ such that for all $i$, if $b_i = 1$, then $c(x_i) > y_i$, and if $b_i = 0$, then $c(x_i) \leq y_i$, i.e. if $b_i = 1$, then $g_c(x_i, y_i) = 0$, and if $b_i = 0$, then $g_c(x_i, y_i) = 1$. Thus, $(x_1, y_1), (x_2, y_2), \ldots, (x_n, y_n)$ are shattered by $\mathcal{C}$.

We then consider the class $\mathcal{C}_\leq$. Consider a sequence $S = (x_1, y_1), \ldots, (x_n, y_n) \in (\mathcal{X} \times [0, 1])^n$ realizable by $\mathcal{C}$. Assuming $\mathcal{C}_\leq$ has a binary compression scheme $(\kappa_b, \rho_b)$, we construct an $\epsilon$-approximate compression scheme as follows. **Compression:** Inflate $S$ to $S_\epsilon$ and apply $\kappa_b$ to $S_\epsilon$. $\kappa_b(S_\epsilon)$ will have at most $f(d_{\mathrm{P}})$ points and $\leq f(d_{\mathrm{P}})$ additional bits. The elements of $\kappa_b(S_\epsilon)$ are of the form $(x, y)$ where $x \in \mathcal{X}$ and $y \in \mathcal{Y}_\epsilon$. To construct $\kappa(S)$, we can encode $\kappa_b(S)$ as follows: For each $(x, y)$ in $\kappa_b(S_\epsilon)$, there is an $x \in \mathcal{X}$ that was inflated to create $(x, y)$ (contributes 1 to the compression size), and one can use $O(\log \frac{1}{\epsilon})$ bits to encode $y$ (since $y \in \{1\} \cup \{c\epsilon : 0 \leq c \leq \lfloor \frac{1}{\epsilon} \rfloor\}$, and $c$ requires $O(\log \frac{1}{\epsilon})$ bits to encode). Thus, the compression size is $O(f(d_{\mathrm{P}}) \log \frac{1}{\epsilon})$. **Reconstruction**: $\kappa(S)$ includes enough information for us to recover $\kappa_b(S_\epsilon)$, so we can apply $\rho_b$ to $\kappa_b(S_\epsilon)$. For each $x$ value, the output as we increase $y$ over the multiples of $\epsilon$ will be 0 for a (possibly empty) contiguous region, and then 1 for a contiguous region up to 1. We can pick any $y$ in the boundary to get a reconstruction that is within $\epsilon$ of the true value.

To analyze compression in the $\ell_p$ setting, notice that the losses are in $[0, 1]$, so in order for the $\ell_p$ loss to be $\leq \epsilon$, it must hold that $\frac{1}{n} \sum_{i=1}^{n} |\rho(\kappa(S))(x_i) - y_i|^p \leq \epsilon$. The left-hand side is upper-bounded by $\max_{i=1}^{n} |\rho(\kappa(S))(x_i) - y_i|^p$, so it is sufficient that $\max_{i=1}^{n} |\rho(\kappa(S))(x_i) - y_i|^p$ to be $\leq \epsilon$. Taking the $p$th root of both sides, it follows that $\max_{i=1}^{n} |\rho(\kappa(S))(x_i) - y_i| \leq \epsilon^{\frac{1}{p}}$, which implies that the $\ell_\infty$ loss must be at most $\epsilon^{\frac{1}{p}}$. We can plug in the compression bound for $\epsilon^{\frac{1}{p}}$-approximate $\ell_\infty$ compression to get a compression bound of $O\left(f(d_{\mathrm{P}}) \log(\frac{1}{\epsilon^{1/p}})\right) = O\left(\frac{f(d_{\mathrm{P}}) \log(\frac{1}{\epsilon})}{p}\right)$. $\blacksquare$

**Open Problem 17** *Suppose all binary concept classes with VC dimension $d_{\mathrm{VC}}$ have a sample compression scheme of size $f(d_{\mathrm{VC}})$. Does every $[0,1]$-valued concept class with a finite fat-shattering dimension (at any scale) admit an $\epsilon$-approximate $\ell_\infty$ compression scheme of size $O(f(\mathrm{fat}_{c\epsilon})\mathrm{polylog}(\frac{1}{\epsilon},\mathrm{fat}_{c\epsilon}))$ for some $c > 0$, where $\mathrm{fat}_\gamma$ is the fat-shattering dimension of the concept class at scale $\gamma$?*

### 4.1. Additional Assumptions: Majority Votes, Proper, and Stable Compression Schemes

By assuming the existence of majority votes, proper, or stable sample compression schemes for binary classification, we can derive stronger results (similarly to Theorem 7 and Theorem 9 in the multiclass setting).

**Theorem 18 (Regression, Approx. Compression, Reductions with Majority Vote Compression)**
*Suppose any binary concept class $\mathcal{C}$ with VC dimension $d_{\mathrm{VC}} < \infty$ has a compression scheme of size $f(d_{\mathrm{VC}})$, which is a proper or a majority vote compression scheme (see Definition 6). Then for any $p \in [1,\infty]$, any class with pseudo-dimension $d_{\mathrm{P}} < \infty$ admits an $\epsilon$-approximate compression scheme, with respect to the $\ell_p$ loss, of size $O(f(d_{\mathrm{P}}))$.*

**Proof** We only prove the result for the $\ell_\infty$ loss. Note that since the losses are in $[0,1]$, the $\ell_p$ loss is upper bounded by the $\ell_\infty$ loss, so the result for $\ell_\infty$ loss will imply the result for the $\ell_p$ loss.

Suppose $\mathcal{C}_\leq$ has a binary compression scheme $(\kappa_b, \rho_b)$ which is either a proper compression scheme or a majority vote compression scheme. Consider $g = \rho_b(T)$ for any $T$ in the image of $\kappa_b$ on the image of any finite realizable $S$. We claim that for all $x \in \mathcal{X}$, there exists an $r \in [0,1]$ such that for $y \in \mathcal{Y}$, if $y < r$, then $g(x,y) = 0$, and if $y \geq r$, $g(x,y) = 1$. If the compression scheme is proper, we can just let $r = c(x)$ and we are done. Consider the case where the compression scheme is a majority vote compression scheme. Suppose the majority reconstruction is taken from some finite $\mathcal{C}'_\leq \subseteq \mathcal{C}$. Fix $x \in \mathcal{X}$, and increase $y$ over $\mathcal{Y}$. $|\{c \in \mathcal{C}{\leq}' : c(x,y) = 1\}|$ is monotonically non-decreasing and equal to $|\mathcal{C}'_\leq|$ for $y = 1$. $|\{c \in \mathcal{C}'_\leq : c(x,y) = 0\}|$ is monotonically non-increasing and equal to $0$ for $y = 1$. Both sets have the same sum over all pairs $(x,y)$. Therefore, as $y$ increases, the majority starts at 0 for a (possibly empty) prefix, becomes 1 eventually for some $y = r$ (possibly at $y = 0$), and remains 1 as $y$ increases.

We can now construct a compression scheme $(\kappa, \rho)$ for $\mathcal{C}$ as follows. **Compression**: Given realizable $S = (x_1, y_1), \ldots, (x_n, y_n)$, consider $S_\epsilon$ (Equation 4). We can input to $\kappa_b$ the set $T$, which for every $1 \leq i \leq n$, contains $((x_i, y), 1)$ for the smallest $y \in \mathcal{Y}_\epsilon$ that is $\geq y_i$, and $((x_i, y), 0)$ for the largest $y \in \mathcal{Y}_\epsilon$ that is $< y_i$ (this may not exist, in which case do not include it in $T$). Applying $\kappa_b$ to $T$ will return $f(d_{\mathrm{P}})$ points in $S_\epsilon$. $\kappa$ can output the $(x_i, y_i)$ pairs responsible for $\kappa_b(T)$, and for each point, use two bits: the first one to indicate if the corresponding $((x,y), 0)$ pair in $T$ was output by $\kappa_b$, and the second one to indicate if the corresponding $((x,y), 1)$ pair in $T$ was output by $\kappa(b)$. We use at most three bits for each point, so the compression size is at most $3f(d_{\mathrm{P}})$. **Reconstruction**: We have shown that the output of $\kappa(S_\epsilon)$ has elements of the form $(x,y)$ and at most three bits per point. We can use this information to recover the points in $\kappa_b(T)$. Then, compute $\rho_b(\kappa_b(T))$, which is guaranteed to be correct on $T$, and has the property that as $y$ increases, it outputs 0 for a (possibly empty) prefix and 1 for a suffix. To find the $\epsilon$-approximate output for an input $x \in \mathcal{X}$, iterate over the elements in $\mathcal{Y}_\epsilon$ in increasing order, and output the first element for which $\rho_b(\kappa_b(x,y)) = 1$. ∎

**Theorem 19 (Regression, Approximate Compression, Reductions with Stable Compression)**
*Suppose any binary concept class $\mathcal{C}$ with VC dimension $d_{\mathrm{VC}} < \infty$ has a stable compression scheme of size $f(d_{\mathrm{VC}})$. Then, for any $p \in [1, \infty]$, any class with pseudo-dimension $d_{\mathrm{P}} < \infty$ admits an $\epsilon$-approximate compression scheme, with respect to $\ell_p$ loss, of size $O(f(d_{\mathrm{P}}))$.*

**Proof** As in Theorem 18, we only prove the result for $\ell_\infty$, and that will imply the result for general $\ell_p$. Assume there is a stable binary compression scheme $(\kappa_b, \rho_b)$ for $\mathcal{C}_\leq$. **Compression:** Inflate $S$ to $S_\epsilon$, and run $\kappa_b$ on $S_\epsilon$. This will return $O(f(d_{\mathrm{P}}))$ points. Since the compression scheme is stable, we convert this to $T_\epsilon$ where $T$ corresponds to the $(x_i, y_i)$ pairs for which $x_i \in \kappa_b(S_\epsilon)$. We can represent $T_\epsilon$ via $T$, giving a compression size $f(d_{\mathrm{P}})$. **Reconstruction:** Reinflate $T$ to $T_\epsilon$, and $\kappa(T_\epsilon)$ gives the desired binary predictor in $\mathcal{C}_\leq$. ■

We are able to compute an $\epsilon$-approximate sample compression scheme with size $O(f(d_{\mathrm{P}}))$, assuming the binary compression scheme is a majority vote, proper, or stable. It is notable that this bound is independent of $\epsilon$, suggesting the possibility of extending similar results to exact compression schemes. The above results lead to the following open problem, which has multiple subproblems:

**Open Problem 20** *Suppose any binary concept class $\mathcal{C}$ with VC dimension $d_{\mathrm{VC}} < \infty$ has a compression scheme of size $f(d_{\mathrm{VC}})$.*

1. *If the binary compression scheme is either proper, majority vote, or stable, does every class with pseudo-dimension $d_{\mathrm{P}} < \infty$ admit an exact compression scheme of size $O(f(d_{\mathrm{P}}))$?*

2. *If there are no assumptions on the binary compression scheme, given $\epsilon > 0$ and $p \in [1, \infty]$, does every class with pseudo-dimension $d_{\mathrm{P}} < \infty$ admit an $\epsilon$-approximate compression scheme of size $O(f(d_{\mathrm{P}}))$ for $\ell_p$ loss?*

3. *If there are no assumptions on the binary compression scheme, does every class with pseudo-dimension $d_{\mathrm{P}} < \infty$ admit an exact compression scheme of size $O(f(d_{\mathrm{P}}))$?*

If we assume the existence of a stable infinitized or inflated compression scheme for $\mathcal{C}_{\mathcal{Y}}$ (see Equation 2), we can establish an exact compression (rather than approximate), similarly to Theorem 11 and Theorem 13 in the multiclass setting. The proofs follow similar reasoning as Theorem 19 and are omitted here for brevity.

**Theorem 21 (Regression, Exact Compression, Infinitized Stable Binary Assumption)**
*Let $\mathcal{C}$ have pseudo-dimension $d_{\mathrm{P}}$. If $\mathcal{C}_{\mathcal{Y}}$ (Equation 2) has an infinitized or inflated stable compression scheme of size $k$, then $\mathcal{C}$ admits a stable (exact) compression scheme of size $k$.*

The following theorem slightly deviates from the theme of earlier proofs about reductions to binary compression. Instead, this is a reduction from realizable regression to multiclass classification with infinite labels, by showing that for any concept class $\mathcal{C} \subseteq [0,1]^{\mathcal{X}}$, it holds that $d_{\mathrm{G}}(\mathcal{C}) \leq 4d_{\mathrm{P}}(\mathcal{C})$. Attias et al. (2024) very briefly stated that there is a bounded realizable exact compression scheme for classes with bounded pseudo-dimension, but we provide the full proof in this paper. The proof is in Appendix C.

**Theorem 22 (Regression, Exact Compression, Reduction to Multiclass with $|\mathcal{Y}| = \infty$)**
*Suppose any multiclass concept class $\mathcal{C}$ with graph dimension $d_{\mathrm{G}} < \infty$ has a compression scheme of size $f(d_{\mathrm{G}})$. Then, any real-valued class with pseudo-dimension $d_{\mathrm{P}} < \infty$ has a compression scheme of size $O(f(4d_{\mathrm{P}}))$. This implies that there is an exact compression scheme of size $O(d_{\mathrm{P}} 2^{4d_{\mathrm{P}}})$.*

**Proof sketch** The idea behind the proof is to consider $n$ points $x_1, x_2, \ldots, x_n$ that are G-shattered by $\mathcal{C}$. We use a pigeonhole argument to prune out concepts from $\mathcal{C}$ to a subset $\mathcal{C}'$ such that there is $y_1, y_2, \ldots, y_n$ where $\{(\mathbb{1}[c(x_1) \leq y_1], \mathbb{1}[c(x_2) \leq y_2], \ldots, \mathbb{1}[c(x_n) \leq y_n] : c \in \mathcal{C}'\}$ is large enough, such that we can apply Sauer's lemma to lower bound the pseudo-dimension.

### 4.2. Agnostic Approximate Compression for Regression

So far, we discussed compression schemes for the realizable case. In the following, we show how to extend it to the agnostic case.

**Theorem 23 (Reducing Agnostic (Approximate) Compression for Regression to Binary)** *Suppose that for binary classes with VC dimension $d_{\mathrm{VC}} < \infty$, there exists a sample compression scheme of size $f(d_{\mathrm{VC}})$. Then, for a real-valued class with pseudo-dimension $d_{\mathrm{P}} < \infty$, there exists an agnostic sample compression scheme with respect to the $\ell_\infty$ loss of size $O(f(d_{\mathrm{P}}) \log(\frac{1}{\epsilon}))$. Furthermore, for $p \in [1, \infty)$, there exists an agnostic sample compression scheme with respect to the $\ell_p$ loss of size $O\left(\frac{f(d_{\mathrm{P}}) \log(\frac{1}{\epsilon})}{p}\right)$.*

**Proof** Consider $S = (x_1, y_1), \ldots, (x_n, y_n)$. Run ERM to compute an $\widehat{c}$ that minimizes the following:

$$\inf_{c \in \mathcal{C}} \max_{1 \leq i \leq n} |c(x_i) - y_i|.$$

Consider the dataset $(x_1, y_1'), \ldots, (x_n, y_n')$, where we define $y_i'$ to be $\widehat{c}(x_i)$. This is realizable by the class $\mathcal{C}_{\leq}$ from the proof of Theorem 16, so we can apply the realizable compression scheme to get a compression size of $O(f(d_{\mathrm{P}}) \log(1/\epsilon))$. To prove the result for general $\ell_p$ loss, we have as in the proof of Theorem 16 that since the losses are in $[0, 1]$, in order for the $\ell_p$ loss to be $\leq \epsilon$, it s sufficient for the $\ell_\infty$ loss to be $\leq \epsilon^{\frac{1}{p}}$. Thus, it s sufficient to get an $\epsilon^{\frac{1}{p}}$-approximate $\ell_\infty$ agnostic compression scheme, and the one that was just constructed will have size $O\left(\frac{f(d_{\mathrm{P}}) \log(\frac{1}{\epsilon})}{p}\right)$. ∎

It remains a major open problem to determine whether there exists an exact agnostic compression scheme for regression with the $\ell_1$ loss of size $O(d_{\mathrm{P}})$, while the best-known result is an $\epsilon$-approximate agnostic compression scheme of size $\frac{1}{\epsilon} 2^{O(\mathrm{fat}_{c\epsilon})}$ for some constant $c > 0$ (see Attias et al. (2024)).

## 5. Compression for Adversarially Robust Classification Against Test-Time Attacks

In this section, we tackle the problem of compression in the adversarially robust classification setting, by reducing it to the binary classification problem. In this setting, there is a perturbation function $\mathcal{U} : \mathcal{X} \to 2^{\mathcal{X}}$ that takes an input and outputs a perturbation set, satisfying $x \in \mathcal{U}(x)$ for all $x \in \mathcal{X}$. A loss is incurred if the input can be perturbed in such a way that the model predicts a label different from the true label. More specifically, for $c \in \mathcal{C}$, $x \in \mathcal{X}$, and $y \in \mathcal{Y}$, the loss

function is $\ell_{\mathcal{U}}(c, x, y) = \mathbb{1}[\exists z \in \mathcal{U}(x) : c(z) \neq y]$ at testing time, i.e., there exists some perturbation $z \in \mathcal{U}(x)$ for which the classifier $c$ predicts a label different from $y$. Several prior works have explored the sample complexity of adversarially robust learning, with Montasser et al. (2019); Attias et al. (2022b) focusing on binary classification and Attias and Hanneke (2023) extending this to real-valued classes.

To achieve compression in this adversarially robust setting, we focus on scenarios where the dataset is robustly realizable, meaning that the dataset can be perfectly labeled by a concept regardless of how the data is perturbed. We formally define this notion below.

**Definition 24 (Robust Realizability)** *Consider a perturbation function $\mathcal{U} : \mathcal{X} \to 2^{\mathcal{X}}$ with $x \in \mathcal{U}(x)$ for all $x \in \mathcal{X}$. Given a binary class $\mathcal{C}$, define $(x_1, y_1), \ldots, (x_n, y_n)$ to be robustly realizable if*

$$\inf_{c \in \mathcal{C}} \frac{1}{n} \sum_{i=1}^{n} \sup_{z \in \mathcal{U}(x_i)} \mathbb{1}[c(z) \neq y_i] = 0.$$

Given the notion of robust realizability, we now define an adversarially robust compression scheme that can compress the dataset while ensuring accuracy on all perturbed examples. For a compression scheme to be adversarially robust, it must select a small subset of the data and return a predictor that remains accurate on all data points, even when the inputs are adversarially perturbed. The goal is to maintain the predictor's accuracy while reducing the amount of stored data.

**Definition 25 (Adversarially Robust Compression Scheme)** *Given a concept class $\mathcal{C}$ and perturbation function $\mathcal{U}$, we say that a sample compression scheme $(\kappa, \rho)$ is adversarially robust if for any*
$(x_1, y_1), \ldots, (x_n, y_n)$ *that is robustly realizable, for all $i = 1, \ldots, n$ $\sup_{z \in \mathcal{U}(x_i)} |\rho(\kappa(S))(z) - y_i| = 0$.*

We prove the following theorem for adversarially robust compression.

**Theorem 26 (Reducing Adversarially Robust Compression to Binary Compression)** *Let $\mathcal{U} : \mathcal{X} \to 2^{\mathcal{X}}$ be a perturbation function and let $M = \sup_{x \in \mathcal{X}} |\mathcal{U}(x)|$ be finite. Suppose any binary concept class $\mathcal{C}$ with VC dimension $d_{\mathrm{VC}} < \infty$ has a sample compression scheme of size $f(d_{\mathrm{VC}})$. Then, $\mathcal{C}$ has an adversarially robust sample compression scheme of size $O(f(d_{\mathrm{VC}}) \log M)$.*

**Proof** Let $S = (x_1, y_1), \ldots, (x_n, y_n)$ be a set that is robustly realizable by a class $\mathcal{C}$. Let $(\kappa_b, \rho_b)$ be a binary compression scheme for $\mathcal{C}$. We construct a compression scheme $(\kappa, \rho)$ as follows. **Compression:** Given $S$, inflate it to create $S_{\mathcal{U}} = \{(z, y_i) : i \in [n], z \in \mathcal{U}(x_i)\}$. Apply $\kappa_b$ to $S_{\mathcal{U}}$ to obtain a compression set of size $f(d_{\mathrm{VC}})$. For each point $(z, y_i)$ in the compression set, we can encode $z$ using $O(\log M)$ bits by storing its index in $\mathcal{U}(x_i)$. Thus, the total compression size is $O(f(d_{\mathrm{VC}}) \log M)$. **Reconstruction:** Apply $\rho_b$ to the compressed set to recover the concept. The recovered concept will be consistent with all points in $S_{\mathcal{U}}$, and thus robustly consistent with $S$. ∎

By assuming the existence of stable sample compression schemes for binary classification, we can derive an adversarial robust compression scheme of size independent of $|\mathcal{U}(x)|$.

**Theorem 27 (Robust Compression Assuming Existence of Stable Binary Compression)**
*Let $\mathcal{U} : \mathcal{X} \to 2^{\mathcal{X}}$ be a perturbation function and let $M = \sup_{x \in \mathcal{X}} |\mathcal{U}(x)|$ be finite. Suppose any binary concept class $\mathcal{C}$ with VC dimension $d_{\mathrm{VC}} < \infty$ has a stable sample compression scheme of size $f(d_{\mathrm{VC}})$. Then, $\mathcal{C}$ has an adversarially robust sample compression scheme of size $O(f(d_{\mathrm{VC}}))$.*

**Proof** Let $S = (x_1, y_1), \ldots, (x_n, y_n)$ be a set that is robustly realizable by a class $\mathcal{C}$. Let $(\kappa_b, \rho_b)$ be a stable binary compression scheme for $\mathcal{C}$. We construct a compression scheme $(\kappa, \rho)$ as follows. **Compression:** Given $S$, inflate it to create $S_{\mathcal{U}} = \{(z, y_i) : i \in [n], z \in \mathcal{U}(x_i)\}$. Apply $\kappa_b$ to $S_{\mathcal{U}}$ to obtain a compression set. By the properties of $\kappa_b$, there exists a set $T \subseteq S$ of size $f(d_{\mathrm{VC}})$ such that all points in $\kappa_b(S_{\mathcal{U}})$ come from inflating $T$. Since the compression scheme is stable, we can output $T$ as our compression, giving a size of $f(d_{\mathrm{VC}})$. **Reconstruction:** Given $T$, inflate it to $T_{\mathcal{U}}$ and apply $\rho_b(\kappa_b(T))$ to reconstruct the concept. Since the binary compression scheme is stable, this will give the same result as applying $\rho_b$ to $\kappa_b(S_{\mathcal{U}})$, ensuring robust consistency with $S$. ∎

**Open Problem 28** *Let $\mathcal{C} \subseteq \{0, 1\}^{\mathcal{X}}$ be a binary concept class with VC dimension $d_{\mathrm{VC}} < \infty$, and let $\mathcal{U} : \mathcal{X} \to 2^{\mathcal{X}}$ be a perturbation function. Suppose $\mathcal{X}$ has a sample compression scheme of size $f(d_{\mathrm{VC}})$. Does there exist an adversarially robust compression scheme of size $O(f(d_{\mathrm{VC}}))$ for $\mathcal{C}$?*

### 5.1. Negative Result for Adversarially Robust Compression

While bounded-size sample compression schemes are known to exist for binary classification problems with classes of finite VC dimension, we present a negative result for the adversarially robust setting. Specifically, we show that there exists a robustly learnable concept class that does not admit any bounded-size sample compression scheme. A similar phenomenon has been observed in multiclass classification (Pabbaraju, 2024) and list learning (Hanneke et al., 2024). The proof is in Appendix D.

**Theorem 29 (Negative Result for Adversarially Robust Compression)** *There exists a concept class which is robustly learnable, but has no bounded-size adversarially robust compression scheme.*

**Proof sketch** To prove this theorem, we consider a partial concept class $\mathcal{C}_{part}$, which has VC dimension 1 but no bounded-size compression scheme, and constructed in Theorem 6 of Alon et al. (2022) (summarized in Lemma 39). Let $\mathcal{C}_{part}$ have domain $\mathcal{X}$, and let each $x \in \mathcal{X}$ have a unique twin $x'$ that lies outside of $\mathcal{X}$, Define $\mathcal{X}' = \{x' : x \in \mathcal{X}\}$, and set $\widetilde{\mathcal{X}}$ as $\mathcal{X} \cup \mathcal{X}'$. We now define a new class $\mathcal{C} \subseteq \{0, 1\}^{\widetilde{\mathcal{X}}}$, where $\mathcal{C} = \{g_c : c \in \mathcal{C}_{part}\}$ and for each $x \in \mathcal{X}$, if $x \in \mathrm{supp}(\mathcal{C}_{part})$, $g_c(x) = g_c(x') = c(x)$, and otherwise, $g_c(x) = 0$ and $g_c(x') = 1$. We define the perturbation function to be $\mathcal{U}(x) = \mathcal{U}(x') = \{x, x'\}$ for all $x \in \mathcal{X}$. One can show that if $\mathcal{C}$ has a bounded-size robust compression scheme with respect to $\mathcal{U}$, this will imply that $\mathcal{C}_{part}$ has a bounded-size compression scheme, which is not possible by Lemma 39. To show that $\mathcal{C}$ is robustly learnable with respect to $\mathcal{U}$, we consider the one-inclusion graph (Definition 41) and show that it has no cycles. This allows the edges to be oriented to have a maximum out-degree 1, and it follows that the class is robustly learnable.

### Acknowledgments

Idan Attias is supported by the National Science Foundation under Grant ECCS-2217023, through the Institute for Data, Econometrics, Algorithms, and Learning (IDEAL).

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

## Appendix A. Discussion about Infinitized Sample Compression Schemes

For binary concept classes with VC dimension $d_{\mathrm{VC}}$, Moran and Yehudayoff (2016) demonstrated the existence of a constant-size sample compression scheme of size $2^{O(d_{\mathrm{VC}})}$. However, we show that when the number of samples is infinite, such a result is no longer possible. Additionally, we construct an infinitized compression scheme for classes with finite Littlestone dimension. However, we also show that this is not a necessary condition—there are cases where the Littlestone dimension is infinite, yet the class still admits an infinitized compression scheme.

**Definition 30 (Littlestone Dimension (Littlestone, 1988))** *A binary tree is perfect if all internal nodes have exactly two children, and all leaf nodes are at the same level. It is said to have depth $d$ if there are $2^d - 1$ vertices. Given a class $\mathcal{C}$, a Littlestone tree is a perfect binary tree, where each vertex is labeled with an element from $\mathcal{X}$. For each path from the root to a leaf, there exists a concept $c \in \mathcal{C}$ such that for each internal node at depth $i$ with label $x_i$, $c(x_i) = b_i$, where $b_i \in \{0, 1\}$ indicates whether the path follows the left ($b_i = 0$) or right ($b_i = 1$) child. The* Littlestone dimension *of $\mathcal{C}$ (denoted as $d_{\mathrm{LD}}(\mathcal{C})$) is the maximum depth of a Littlestone tree of $\mathcal{C}$. If no largest depth exists, set $d_{\mathrm{LD}}(\mathcal{C}) = \infty$.*

Note that an infinite Littlestone tree implies an infinite Littlestone dimension, but the converse is not true. A class can have an infinite Littlestone dimension (meaning that for any depth, there exists a Littlestone tree) without having an infinite Littlestone tree.

For a concept class $\mathcal{C}$, let $\mathcal{C}_S$ be the version space with respect to $S$, i.e., $\{c \in \mathcal{C} : c(x) = y$ for all $(x, y) \in S\}$. When the concept class $\mathcal{C}$ is clear from the context, we use $V_S$ to denote it, and let $V_{S,x,y} := V_{S \cup \{(x,y)\}}$. Additionally, we introduce the following oracle $M(f, S)$. For a function $f : \mathcal{X} \to \mathcal{Y}$ and a (possibly infinite) sample set $S \subseteq \mathcal{X} \times \mathcal{Y}$, the oracle $M(f, S)$ returns true if there exists an $(x, y) \in S$ such that $f(x) \neq y$, and false otherwise. This oracle allows us to determine whether a predictor makes any mistakes on the infinite sample set $S$.

**Theorem 31 (Infinitized Compression Scheme for Littlestone Classes)** *Let a binary concept class $\mathcal{C} \subseteq \{0,1\}^{\mathcal{X}}$ with a finite Littlestone dimension $d_{\mathrm{LD}}(\mathcal{C}) < \infty$. Then, assuming access to the oracle $M(\cdot, \cdot)$, there exists an infinitized sample compression scheme (Algorithm 1) of size $O(d_{\mathrm{LD}}(\mathcal{C}))$ for $\mathcal{C}$.*

The idea for the sample compression scheme is closely related to Littlestone's *Standard Optimal Algorithm* (SOA) (Littlestone, 1988).

**Proof** We construct the compression $(\kappa, \rho)$ as follows. Suppose we are given a set $S$ of $(x, y)$ pairs (possibly infinite) that is realizable by $\mathcal{C}$. We construct $\kappa(S)$ via the following process in Algorithm 1. To prove the compression size, we claim the algorithm will take $d_{\mathrm{LD}}(\mathcal{C})$ iterations. Whenever a mistake is made for some $(x, y)$ at time $t$, we must have that $d_{\mathrm{LD}}(V_{T_{t+1}}) < d_{\mathrm{LD}}(V_{T_t})$, since otherwise, $d_{\mathrm{LD}}(V_{T_t,x,0}) = d_{\mathrm{LD}}(V_{T_t,x,1}) = d_{\mathrm{LD}}(V_{T_t}) = k$ for some $k$, so we can shatter $V_{T_t}$ using $x$ at the root and the Littlestone trees of depth $k$ for each labeling of $x$, implying a larger Littlestone dimension for $V_{T_t}$. Thus, the algorithm takes at most $d_{\mathrm{LD}}(\mathcal{C})$ iterations and we have an infinitized compression scheme of size $O(d_{\mathrm{LD}}(\mathcal{C}))$. ∎

While finite Littlestone Dimension implies an infinitized compression, the opposite direction does not hold – consider the following example:

---

**Algorithm 1** Infinitized Compression SOA

---

**Input:** Concept class $\mathcal{C} \subseteq \{0,1\}^{\mathcal{X}}$, $S = \{(x_i, y_i) : i \in \mathcal{I}\}$ for some (possibly infinite) index set $\mathcal{I}$.
**Initialize:**

- Initial compression set $T_0 \leftarrow \varnothing$.

- Let $\rho : 2^{\mathcal{X}} \times \mathcal{X} \to \{0,1\}$, define $\rho(T)(x)$ as follows:

    - If $(x, y) \in T$ for some $y$, then predict $y$.
    - If there exists a fixed $y$ such that $c(x) = y$ for all $c \in \mathcal{C}_T$, predict $y$.
    - Predict according to $\arg \max_{y \in \{0,1\}} d_{\mathrm{LD}}(V_{T,x,y})$, and predict 1 if there is a tie.

For $t = 0, 1, \ldots$:

1. If the predictor $\rho(T_t)$ is wrong for any $(x, y) \in S$ (determined using the oracle query $M(\rho(T_t), S)$), then let $T_{t+1} = T_t \cup \{(x, y)\}$.

2. Otherwise ($\rho(T_t)$ does not make any mistake on $S$) return the compression $\kappa(S) = T_t$.

---

**Example 2 (A Class with Infinite Littlestone Dimension and Infinitized Compression Scheme)**
*Consider the thresholds on the natural numbers. Here $\mathcal{X} = \mathbb{N}$, and $\mathcal{C} \subseteq \{0,1\}^{\mathcal{X}}$ where $\mathcal{C} = \{c_t : t \in \mathbb{N}\}$ and $c_t(x) = \mathbb{1}[x \leq t]$. This class has an infinite Littlestone dimension. It also has a sample compression scheme of size 2 – pick the rightmost point with a label equal to 0 and the leftmost point with a label equal to 1.*

Additionally, there is a class of VC dimension 1 that has no infinitized compression scheme, namely the thresholds over the real numbers.

**Example 3 (VC Class with No Infinitized Compression Scheme)** *Consider the class of thresholds, $\mathcal{C} = \{c_t : t \in \mathbb{R}\}$ where $c_t(x) = \mathbb{1}[x \leq t]$. There is no infinitized compression scheme for this class.*

*We show there is no infinitized compression scheme for $\mathcal{C}$. We can assume for now that $\mathcal{C} = \{c_t : t \in \mathbb{R} \setminus \mathbb{Q}\}$. Let $S = \mathbb{Q}$. Assume there is an compression scheme $(\kappa, \rho)$ for the class where $|\kappa| \leq k$. Notice that for any $t_1, t_2$ irrational, there exists a rational $z$ such that $t_1 < z < t_2$. Thus, $\kappa(S)$ must be different for every $c_t \in \mathcal{C}$. Thus, there must be a one-to-one function from $\mathbb{R} \setminus \mathbb{Q}$ to $\cup_{i \leq k}(\{0,1\} \times S)^i$. However, the former set is uncountable, while the latter set is countable, so this is impossible.*

Notice that the class $\mathcal{C}$ from Example 3 has an infinite Littlestone tree, which naturally leads one to ask whether classes with an infinite Littlestone tree do not have infinitized compression. Note that the class in Example 2 has an infinite Littlestone dimension but not an infinite Littlestone tree. Next, we give an example of a class with an infinite Littlestone tree and a bounded-size compression scheme.

**Example 4 (A Class with Infinite Littlestone Tree and Infinitized Compression Scheme)** *We construct a concept class $\mathcal{C}$ that has an infinite Littlestone tree (and thus infinite Littlestone dimension) yet admits a simple infinitized compression scheme of size 1.*

*Consider an infinite domain $\mathcal{X}$ arranged as an infinite perfect binary tree. For each node $u$ at a finite depth in the tree, let $P_u = x_1, x_2, \ldots, x_k$ be the path from the root to $u$. Define the concept $c_u$ as follows:*

$$c_u(x) = \begin{cases} 1 & x \text{ has a right child in } P_u \\ 0 & \text{otherwise} \end{cases}$$

*If $u$ is a left child, $c_u$ equals $c_v$ where $v$ is $u$'s parent, so we only need concepts for right children. Let $\mathcal{C} = \{c_u : u \text{ is the root or is a right child at finite depth}\}$. This class has an infinite Littlestone tree since we can construct a prediction tree of arbitrary depth by following paths down the binary tree. However, $\mathcal{C}$ admits a simple compression scheme of size 1: Given a sample $S$, if no points are labeled 1, output $\varnothing$. Otherwise, output the deepest point $u$ with label 1, and let the reconstruction predict according to $c_u$.*

It is an interesting open problem to characterize when infinitized compression is possible.

## Appendix B. A Sample Compression Scheme for Classes with Graph Dimension 1

**Lemma 32 (Sample Compression Scheme for Graph Dimension 1)** *Any concept class $\mathcal{C}$ with graph dimension 1 admits a sample compression scheme of size 1.*

Before proving Lemma 32, we summarize some relevant results from Ben-David (2015).

**Definition 33** *A partial ordering $\leq$ over a set $\mathcal{X}$ is called a "tree ordering" whenever for all $x \in \mathcal{X}$, $I_x = \{y : x \leq y\}$ is a linear ordering. Additionally, given a totally ordered set under $\leq$, define the* deepest *element to be the one that is less than or equal to all the others under the ordering $\leq$.*

The following lemma follows from Lemma 4 and Theorem 5 from Ben-David (2015).

**Lemma 34 (Tree Orderings for Classes with VC Dimension 1 (Ben-David, 2015))** *Consider a binary class $\mathcal{C}$ on $\mathcal{X}$ with $d_{\mathrm{VC}}(\mathcal{C}) \leq 1$. Pick any $c_0 \in \mathcal{C}$. Define a partial ordering $\leq$ on $\mathcal{X}$ as follows: for $x, y \in \mathcal{X}$, let $x \leq y$ if, for every $c \in \mathcal{C}$, $c(x) \neq c_0(x)$ implies $c(y) \neq c_0(y)$. It holds that $\leq$ is a tree ordering. Furthermore, for any $c \in \mathcal{C}$, $\{x : c(x) \neq c_0(x)\}$ is a linear ordering.*

Ben-David (2015) provides the following sample compression scheme for classes with VC dimension 1. Consider $S = (x_1, y_1), \ldots, (x_n, y_n)$ realizable by some $c \in \mathcal{C}$. We can pick a $c_0 \in \mathcal{C}$ and consider the ordering $\leq$ from Lemma 34. Identify the deepest point $x_i \in \{x \in \mathcal{X} : c(x) \neq c_0(x)\}$. The compression set consists of the single point $(x_i, y_i)$. For any test point $z$, if there exists exactly one possible label value among concepts consistent with $(x_i, y_i)$ (that is, if $|\{c(z) : c(x_i) = y_i, c \in \mathcal{C}\}| = 1$), predict that unique value. Otherwise, default to predicting according to $c_0$.

We introduce some notation and an algorithm that will be used to prove Lemma 32. Consider a class $\mathcal{C}$ for which $d_{\mathrm{G}}(\mathcal{C}) = 1$. Pick any $c_0 \in \mathcal{C}$. For $c \in \mathcal{C}$, define $g_c$ such that $g_c(x) = \mathbb{1}[c(x) \neq c_0(x)]$. Note the $g_{c_0}(x) = 0$ for all $x$. Let $\mathcal{C}' = \{g_c : c \in \mathcal{C}\}$. $\mathcal{C}'$ is a binary class with VC dimension $\leq 1$. Consider points $x_1, \ldots, x_n$. We can construct the tree ordering from Lemma 34 with respect to $g_{c_0}$ on these points, where for any $x \leq y$ $g_c(x) \neq g_{c_0(x)}$ implies $g_c(y) \neq g_{c_0}(y)$. Since $g_{c_0}(x)$ is zero for all $x$, this implies that whenever $g_c(x) = 1$, it must also hold that $g_c(y) = 1$. By Lemma 34, we have that for every $c \in \mathcal{C}$, the set $\{x : g_c(x) = 1\}$ is a linear ordering. Below, we'll propose what one may consider a natural first attempt at a compression scheme for multiclass, based on the binary compression scheme from Ben-David (2015).

**An attempt at a compression scheme:** Consider the following attempt at a compression scheme, given points $(x_1, y_1), (x_2, y_2), \ldots, (x_n, y_n)$ realizable by some $c \in \mathcal{C}$. Construct the tree from earlier (constructed from some $c_0 \in \mathcal{C}$), and compress to the deepest point $x$ where $g_c(x) = 1$. For a test point $z$, we can try doing the same as earlier: If there are no ambiguities, predict the only option. Otherwise, predict according to $c_0$. However, there are some caveats.

For a test point $z$, if there are no ambiguities, then we are done. Consider the scenario where there are ambiguities. In the binary case, if $x \leq z$, no ambiguities arise as $c(z) \neq c_0(z)$. However, for multiclass prediction, additional ambiguities may occur for points $z$ with $x \leq z$. Below, we will provide a way to address this issue.

**A minor fix to the compression scheme:** We can compress via Algorithm 2. The following

---

**Algorithm 2** Sample Compression Scheme for Classes with Graph Dimension 1

**Input:** Concept class $\mathcal{C} \subseteq \mathcal{Y}^{\mathcal{X}}$, $d_G(\mathcal{C}) \leq 1$, realizable $S = (x_1, y_1), (x_2, y_2), \ldots, (x_n, y_n)$ via some $c \in \mathcal{C}$.

**Initialize:**

- Construct tree ordering $\leq$ from $\{x_1, \ldots, x_n\}$ (As in Lemma 34 statement, from some $c_0 \in \mathcal{C}$).

- $z_1$: Deepest point $x$ such that $g_c(x) = 1$

For $t = 1, 2, \ldots$:

- Let $S_t = \{z : z_t \leq z, |c'(z) : c' \in \mathcal{C} \text{ and } c'(z_t) = c(z_t)| > 1\}$

  - If $S_t = \varnothing$, return $\kappa(S) = \{(z_t, c(z_t)\}$
  - Else let $z_{t+1}$ be the deepest element in $S_t$.

For a test point $z$, define $\rho(\{x, y\})(z)$ to be

$$\rho(\{x, y\})(z) = \begin{cases} h(z) \text{ for any } h \in \mathcal{C} \text{ with } h(x) = y & \text{if } |\{h'(z) : h' \in \mathcal{C} \text{ and } h'(x) = y\}| = 1 \\ c_0(z) & \text{otherwise.} \end{cases}$$

i.e. if there are no ambiguities, predict according to any concept consistent with $(x, y)$, and predict according to $c_0$ otherwise.

---

Lemma suffices to prove the correctness of the algorithm.

**Lemma 35** *In Algorithm 2, when setting $z_{t+1}$ to be the deepest element in $S_t$, $\{c' \in \mathcal{C} : c'(z_{t+1}) = c(z_{t+1})\} \subseteq \{c' \in \mathcal{C} : c'(z_t) = c(z_t)\}$.*

**Proof** Consider the undirected bipartite graph on $\{z_t, z_{t+1}\} \times \mathcal{Y}$, where there is an edge between $(z_t, i)$ and $(z_{t+1}, j)$ whenever there exists an $c' \in \mathcal{C}$ such that $c'(z_t) = i, c'(z_{t+1}) = j$.

Since the $d_G(\mathcal{C}) \leq 1$, the graph must be acyclic (Otherwise, consider a cycle. Since the graph is bipartite, the length of the cycle is at least 4. Picking two disjoint edges $e_1 = ((z_t, i), (z_{t+1}, j)), e_2$, we can G-shatter $z_t$ and $z_{t+1}$ with labels $i$ and $j$ using $e_1, e_2$, and the two neighbors of $e_1$, contradicting that $d_G(\mathcal{C}) \leq 1$. Thus, we can consider the graph to be an undirected forest. Furthermore, for any two disjoint edges, both of them must be incident to a leaf. (Otherwise, if there are two

disjoint edges $e_1, e_2$ that are disjoint, and $e_1$ is not incident with a leaf, then we can shatter the two points as earlier using the labels corresponding to $e_1$ with $e_1$, and the two neighbors of $e_1$).

Now, consider $z_t$ and $z_{t+1}$. Since both of them are on $I_x$ from the tree ordering, it must be the case that $c_0(z_t) \neq c(z_t)$ and $c_0(z_{t+1}) \neq c(z_{t+1})$. Thus, $c$ and $c_0$ will be disjoint in the above bipartite graph, and thus, $c$ must be incident to a leaf. Since $z_{t+1} \in S_t$, $(z_t, c(z_t))$ must lie on an internal node. Thus, $(z_{t+1}, c(z_{t+1}))$ is a leaf. Thus, when we switch the compression point to $z_{t+1}$, $\{c' \in \mathcal{C} : c'(z_{t+1}) = c(z_{t+1})\} \subseteq \{c' \in \mathcal{C} : c'(z_t) = c(z_t)\}$. ∎

**Proof** [of Lemma 32] Utilizing Lemma 35, we now proceed with the proof of this lemma. Consider running Algorithm 2. Each time we switch our compression point from $z_t$ to $z_{t+1}$, by Lemma 35, the space of hypotheses consistent with the compression point becomes a strict subset of what it was before. Furthermore, $z_{t+1} \notin S_{t+1}$, and since $S_{t+1} \subseteq S_t$, $S_t$ keeps shrinking as we increase $t$. All points in $S_t$ lie in $I_{z_1}$ and are larger than $z_t$ in the tree ordering. Since $I_{z_1}$ is finite and $S_t$ keeps shrinking, this process must end in a finite number of steps when $S_t$ becomes empty, at which point there will be no ambiguities in predicting labels for any test point $z \in I_{z_1}$. ∎

## Appendix C. Exact Compression for Realizable Regression: Proof of Theorem 22

We start with the following Lemma, which will relate the graph dimension and the pseudo-dimension.

**Lemma 36 (Graph Dimension Upper Bound via Pseudo-Dimension)** *For any concept class* $\mathcal{C} \subseteq [0, 1]^{\mathcal{X}}$, *it holds that* $d_{\mathrm{G}}(\mathcal{C}) \leq 4d_{\mathrm{P}}(\mathcal{C})$.

**Proof** Consider $n$ points $S_n = \{x_1, \ldots, x_n\}$ that are G-shattered (see Definition 3) by $\mathcal{C}$ via values $y_1, \ldots,$
$y_n$, via $2^n$ concepts $\mathcal{C}_0 \subseteq \mathcal{C}$. Consider $\mathcal{C}_0$ supported on $x_1, \ldots, x_n$. We will consider $d_{\mathrm{P}}(\mathcal{C}_0)$, which will be a lower bound for $d_{\mathrm{P}}(\mathcal{C})$. For any $c \in \mathcal{C}_0$, define

$$g_c(x_i) = \begin{cases} 1 & c(x_i) > y_i, \\ 0 & c(x_i) = y_i, \\ -1 & c(x_i) < y_i. \end{cases}$$

We will prune concepts to form a sequence $\mathcal{C}_0 \supseteq \mathcal{C}_1 \supseteq \ldots \supseteq \mathcal{C}_n$, all supported on $x_1, \ldots, x_n$. For $i = 1, \ldots, n$, given $\mathcal{C}_{i-1}$, we can prune out the concepts $c$ for which $g_c(x_i)$ occurs the least frequently, i.e. we can do the following:

- Consider $\widehat{z} = \arg\min_z |\{c \in \mathcal{C}_{i-1} : g_c(x_i) = z\}|$.

- Set $\mathcal{C}_i = \mathcal{C}_{i-1} - \{c \in \mathcal{C}_{i-1} : g_c(x_i) = \widehat{z}\}$

It now suffices to bound $d_{\mathrm{P}}(\mathcal{C}_n)$. Let $S_i := \{g_c(x_i) : c \in \mathcal{C}_n\}$. $|S_i| \leq 2$ for all $i$, since we pruned the concepts with the least frequent $g_c(x_i)$ earlier. Thus, we can construct $z_i$ as follows for $1 \leq i \leq n$:

$$z_i = \begin{cases} y_i & 0 \notin S_i \text{ or } |S_i| = 1, \\ y_i + \epsilon \text{ for some } \epsilon < \inf_{c \in \mathcal{C}_n, c(x_i) > y_i} c(x_i) - y_i & -1 \notin S_i, \\ y_i - \epsilon \text{ for some } \epsilon < \inf_{c \in \mathcal{C}_n, c(x_i) < y_i} y_i - c(x_i) & 1 \notin S_i. \end{cases}$$

Note that $(\mathbb{1}[c(x_1) \geq z_1], \mathbb{1}[c(x_2) \geq z_2], \ldots, \mathbb{1}[c(x_n) \geq z_n])$ is distinct for all $c \in \mathcal{C}_n$, so it suffices to bound $|\mathcal{C}_n|$ and apply Sauer's Lemma (Sauer, 1972; Vapnik and Chervonenkis, 1971), as follows: By the construction of $\mathcal{C}_i$ for each $i$, since we remove the smallest set in a partition into three at each step, it follows that $|\mathcal{C}_i| \geq \frac{2}{3}|\mathcal{C}_{i-1}|$. Thus, $|\mathcal{C}_n| \geq 2^n(2/3)^n$ Let $d_P := d_P(\mathcal{C}_n)$. By Sauer's lemma (taking logs), we have that

$$d_P \ln(en/d_P) \geq n(\ln 4/3).$$

We have $d_P \geq 1$ (since by definition, we shatter each $(x_i, z_i)$ pair), so the left-hand side can be upper bounded by $d_P \ln(en) = d_P(1 + \ln n)$. Thus, we have that $d_P \geq n\frac{\ln 4/3}{1+\ln n} \geq \frac{1}{4}n$. ∎

**Proof** [of Theorem 22] Applying Theorem 36 gives that $d_G(\mathcal{C}) \leq 4d_P(\mathcal{C})$, i.e., the graph dimension of $\mathcal{C}$ is at most $4d_P$. By our assumption, there exists a compression scheme of size $f(d_G)$, so this implies that there exists a compression scheme of size $f(4d_P)$, as desired. The well-known bound of David et al. (2016) states that there is a bounded sample compression scheme of size $O(d_G(\mathcal{C})2^{d_G(\mathcal{C})})$. Using the reduction from pseudo-dimension to graph dimension, where $f(x) = cx2^x$ for some constant $c > 0$, we get that there is a compression scheme of size $O(d_P 2^{4d_P})$. ∎

## Appendix D. Existence of Robustly Learnable Class with No Bounded Size Adversarially Robust Compression Scheme: Proof of Theorem 29

To prove the theorem, we utilize a partial concept class from Alon et al. (2022), which has an unbounded compression size and VC dimension equal to 1. We begin with some key definitions.

**Definition 37 (Partial concept classes (Alon et al., 2022))** *A* partial concept class *is a class of concepts* $\mathcal{C} \subseteq \{0, 1, *\}^{\mathcal{X}}$, *which consists of* partial concepts $c : \mathcal{X} \to \{0, 1, *\}$ *For a* $c \in \mathcal{C}$, *define* $\text{supp}(c) = \{x \in \mathcal{X} : c(x) \neq *\}$, *which is the set of points where* $c$ *is defined. A dataset* $(x_1, y_1), \ldots, (x_n, y_n)$ *is* realizable *if there exists a* $c \in \mathcal{C}$ *such that* $x_i \in supp(c)$ *for all* $i$, *and* $c(x_i) = y_i$ *for all* $i$. *A partial concept class* $\mathcal{C}$ shatters *a set* $x_1, \ldots, x_n$ *if*

$$\{(c(x_1), c(x_2), \ldots, c(x_n)) : x_1, x_2, \ldots, x_n \in \text{supp}(c)\} = \{0, 1\}^n.$$

*The VC dimension of a partial concept class is the largest nonnegative integer* $n$ *for which there exist* $x_1, x_2, \ldots, x_n$ *shattered by* $\mathcal{C}$.

**Definition 38 (Partial concept class compression scheme)** *Given a class* $\mathcal{C}$, *a compression scheme is a partial concept class compression scheme if for any sequence* $S$ *realizable by* $\mathcal{C}$, $\rho(\kappa(S))(x) = y$ *for all* $(x, y) \in S$.

The partial concept class that we will use for our negative result for adversarial robustness is summarized via the following Lemma, which was proved in Theorem 6 of Alon et al. (2022). This Lemma constructs a partial concept class with VC dimension 1 that does not admit any bounded-size sample compression scheme.

**Lemma 39 (Partial Concept Class with VC 1 and Unbounded Compression (Alon et al., 2022))**
*There exists a partial concept class $\mathcal{C}_{part}$, such that $d_{\mathrm{VC}}(\mathcal{C}_{part}) = 1$, but there is no bounded-size sample compression scheme for $\mathcal{C}_{part}$.*

In the proof of Theorem 29, we make use of the following class $\mathcal{C}$ and perturbation set function $\mathcal{U}$: Let each $x \in \mathcal{X}$ have a unique twin $x'$ outside of $\mathcal{X}$. Define $\mathcal{X}'$ to be $\{x' : x \in \mathcal{X}\}$, and $\widetilde{\mathcal{X}}$ to be $\mathcal{X} \cup \mathcal{X}'$. Construct $\mathcal{C}$ as follows: $\mathcal{C} = \{g_c : c \in \mathcal{C}_{part}\}$ where for $x \in \mathcal{X}$, if $x \in \mathrm{supp}(c)$, $g_c(x) = g_c(x') = c(x)$, and for $x \notin \mathrm{supp}(c)$, $g_c(x) = 0$ and $g_c(x') = 1$. For $x \in \mathcal{X}$, let $\mathcal{U}(x) = \mathcal{U}(x') = \{x, x'\}$.

**Definition 40 (Adversarially Robust Learnability)**
*Given a perturbation function $\mathcal{U}$, a class $\mathcal{C}$ is robustly learnable if, for any $\epsilon, \delta > 0$ and any robustly realizable distribution $\mathcal{P}$, there exists $n = \mathrm{poly}(\frac{1}{\delta}, \frac{1}{\epsilon})$ [3], such that, given $n$ i.i.d. samples from $\mathcal{P}$, there is an algorithm that returns a hypothesis $\widehat{c} \in \mathcal{Y}^{\mathcal{X}}$ satisfying*

$$\mathbb{P}_{(x,y) \sim \mathcal{P}} \left[ \exists z \in \mathcal{U}(x) : \widehat{c}(z) \neq y \right] < \epsilon$$

*with probability at least $1 - \delta$ over the training sample.*

To show that the class is learnable, we use the one-inclusion graph predictor, which we define as follows. Note that since we are in the robust setting, the definition is slightly different, since we require the vertices to correspond to robustly realizable sequences.

**Definition 41 (One-Inclusion Graph Predictor (Haussler et al., 1994))** *Given a concept class $\mathcal{C} \subseteq \mathcal{Y}^{\mathcal{X}}$ and perturbation function $\mathcal{U}$, the One-Inclusion Graph Predictor is an algorithm $\mathcal{A} : (\mathcal{X} \times \mathcal{Y})^* \to \{0,1\}^{\mathcal{X}}$, defined as follows: Given a dataset $(x_1, y_1), \ldots, (x_n, y_n)$, and test point $z$ consider the following graph: Let the vertices $V$ be*

$$\{(c(x_1), \ldots, c(x_n), c(z)) : c \in \mathcal{C}, (x_1, c(x_1)), \ldots, (x_n, c(x_n)), (z, c(z)) \text{ robustly realizable by } \mathcal{C}\}.$$

*For any two vertices $\mathbf{u} = (u_1, u_2, \ldots, u_n, u_z), \mathbf{v} = (v_1, v_2, \ldots, v_n, v_z)$, there will be an edge between $\mathbf{u}$ and $\mathbf{v}$ if there exists exactly one $i$ such that $u_i \neq v_i$.*

*Orient the edges in the graph to minimize the maximum out-degree, and predict $\widehat{c}(z)$ to be*

$$\widehat{c}(z) = \begin{cases} w \in \{0,1\} & \text{if there exists an edge oriented from } (y_1, \ldots, y_n, 1-w) \text{ to } (y_1, \ldots, y_n, w) \\ w \in \{0,1\} & \text{if } (y_1, \ldots, y_n, w) \in V \text{ and } (y_1, \ldots, y_n, 1-w) \notin V. \end{cases}$$

*The predictor can be assumed to have the same orientation, no matter how the vertices are permuted.*

We now prove the following Lemmas to show the learnability of $\mathcal{C}$.

**Lemma 42 (Cycle-free One-Inclusion Graph)** *Consider the one-inclusion graph, applied to robustly realizable subsets of $\mathcal{X} \times [0, 1]$. This graph has no cycles.*

---

3. There are many definitions of learnability, and it is standard to let $n = \mathrm{poly}(d_{d_{\mathrm{VC}}}(\mathcal{C}), \log \frac{1}{\delta}, \frac{1}{\epsilon})$. We adapt the definition here since our concept class has infinite VC dimension, and we relax the $\log \frac{1}{\delta}$ to $\frac{1}{\delta}$.

**Proof** Suppose the graph has a cycle. For each edge in the cycle (if we traverse the cycle), the label of some $x_i$ will get flipped. Let $i_1, \ldots, i_k$ be the sequence of indices that are flipped when we traverse the cycle, Let $i_{\ell_1} = i_{\ell_2} = a$, with $\ell_1 \neq \ell_2$, $(\ell_2 - \ell_1 + k) \mod k$ minimal. Since the difference is minimal, traversing indices from $\ell_1 + 1$ to $\ell_2 - 1$ (modulo $k$) will traverse through distinct elements. Let one of these elements be $b$. Since this is a cycle, each element that is flipped needs to be flipped at least twice, so there exists an element traversing from $\ell_2 + 1 \ldots \ell_1 - 1$ that is also equal to $b$. This would imply that a subsequence that is equal to $a, b, a, b$, so points $a$ and $b$ can be shattered by $\mathcal{C}_{part}$, contradicting that $d_{\mathrm{VC}}(\mathcal{C}_{part}) = 1$. ∎

**Lemma 43 (Robust learnability over $\mathcal{X} \times \{0, 1\}$)** *$\mathcal{C}$ is robustly learnable over $\mathcal{X} \times \{0, 1\}$.*

**Proof** Let $\mathcal{P}$ be a realizable distribution over $\mathcal{X} \times \{0, 1\}$. Given a realizable dataset $(x_1, y_1), (x_2, y_2),$ $\ldots, (x_n, y_n)$ of size $n$ sampled i.i.d. from $\mathcal{P}$, the error over $\mathcal{P}$ can be expressed as the expected error over $(x_{n+1}, y_{n+1}) \sim \mathcal{P}$ of the One-Inclusion Graph algorithm on $x_1, x_2, \ldots, x_n$. For $i = 1, \ldots, n + 1$, let $\widehat{c}_{-i}$ be the predictor returned by the one-inclusion graph algorithm on points $x_1, x_2, \ldots, x_{i-1}, x_{i+1}, x_{i+2}, \ldots, x_n$. Let $G$ be the oriented one-inclusion graph over $x_1, x_2, \ldots, x_n$ with respect to $\mathcal{C}$, and for each vertex $u$, let $\mathrm{outdeg}_G(u)$ be the out-degree of $u$ in $G$. Let $E(G) \overset{\mathrm{def}}{=} \{(u, v) : u$ is directed towards $v$ in $G\}$. The expected error can be expressed as

$$\mathbb{E}_{(x_1, y_1), \ldots, (x_n, y_n) \sim \mathcal{P}, (x_{n+1}, y_{n+1}) \sim \mathcal{P}} \left[ \mathbb{1}\big[\widehat{c}_{-(n+1)}(x_{n+1}) \neq y_{n+1}\big] \right]$$

$$= \mathbb{E}_{(x_1, y_1), \ldots, (x_{n+1}, y_{n+1}) \sim \mathcal{P}, i \sim [n+1]\}} \left[ \mathbb{1}\big[\widehat{c}_{-i}(x_i) \neq y_i\big] \right]$$

$$= \mathbb{E}_{(x_1, y_1), \ldots, (x_{n+1}, y_{n+1}) \sim \mathcal{P}, i \sim [n+1]} \left[ \mathbb{1}\left[ \Big( (y_1, .., y_i, .., y_{n+1}), (y_1, .., 1 - y_i, .., y_{n+1}) \Big) \in E(G) \right] \right]$$

$$= \mathbb{E}_{(x_1, y_1), \ldots, (x_{n+1}, y_{n+1}) \sim \mathcal{P}} \left[ \frac{\mathrm{outdeg}_G((y_1, \ldots, y_i, \ldots, y_{n+1}))}{n} \right].$$

By Lemma 42, the graph is acyclic. We can orient the edges to have maximum out-degree 1 (the graph is a forest, so we can root each component at an arbitrary node, and direct all the edges downwards). Thus, the expected error is at most $\frac{1}{n}$. Applying Markov's inequality on the error gives that the class is robustly learnable. ∎

**Lemma 44 (Robust learnability over $\widetilde{\mathcal{X}} \times \{0, 1\}$)** *$\mathcal{C}$ is robustly learnable over $\widetilde{\mathcal{X}} \times \{0, 1\}$.*

**Proof** Consider a robustly realizable distribution $\mathcal{P}$ over $\widetilde{\mathcal{X}} \times \{0, 1\}$. Notice that sampling $(x, y) \sim \mathcal{P}$ and then changing the label from $z'$ to $z$ if $x = z'$ for some $z \in \mathcal{X}$ corresponds to sampling from a robustly realizable distribution from $\mathcal{X} \times \{0, 1\}$. Thus, we can convert all the points to be in $\mathcal{X} \times \{0, 1\}$, and directly apply the algorithm from Lemma 43 to get the same error. ∎

**Proof** [of Theorem 29] Consider the class $\mathcal{C}$ from above. First, we will show that $\mathcal{C}$ has no bounded compression scheme. Suppose there is an adversarially robust compression scheme for $\mathcal{C}$ with compression function $\kappa$ and reconstruction function $\rho$, with size $k$. We can notice that $\rho, \kappa$ are also a valid compression scheme for $\mathcal{C}_{part}$, since any dataset that is realizable in $\mathcal{C}_{part}$ is also robustly

realizable. Thus, we have a bounded-size compression scheme for $\mathcal{C}_{part}$, which is a contradiction to Lemma 39. Now, it remains to show that $\mathcal{C}$ is robustly learnable. This is already true by Lemma 44, completing the proof of the theorem.

∎

