# OpenReview forum: "Sample Compression Scheme Reductions"
_algorithmiclearningtheory.org/ALT/2025/Conference — ALT 2025_

### Official Review · Reviewer_b5wq · 2024-11-08

**Rating:** 8
**Confidence:** 1

**Review:**

This paper studies compression in the context of learning theory. Compression refers to encoding a subset of examples along with a reconstruction bit-string, so that the hypothesis on the entire training set can be recovered using just the compressed information. The sample compression conjecture posits that any binary concept class with VC dimension d_VC can be compressed with a size of O(d_VC). While the conjecture remains a significant open problem in learning theory, this paper lays out a series of implications. These come in three categories: (1) Multiclass classification; (2) regression, and (3) adversarially robust classifications. All of the results are derived by reduction to the binary case from the conjecture, constructing a compression scheme by assuming the existence of a compression scheme for the binary setting.
The results are significant, and the paper is well-written.
Typos:
(pg.6) "we have that (x,z) = (x_i, y_i) for some i --> I think it should be (x,y)
(pg. 7) "donclude"

It would be nice to include a discussion of other learning problems (if any) that may be reducible to the binary case. Also, do any of your results cast doubts on the conjecture? (i.e. if A is reduced to B but A cannot be compressed then B cannot be compressed.)

**Paper Award:**

No

---

> ### Author Response · Authors · 2024-11-25
> **Thank you for the review.**
>
> We thank the reviewer for their positive and thoughtful feedback, as well as the insightful suggestions.
>
> Thank you for catching the typos, we will make the necessary corrections.
>
> Regarding the suggestion to discuss other learning problems that may be reducible to the binary case: this is a great idea. We will include such a discussion, specifically addressing whether the setting of distribution learning can be reduced to a binary classification setting.
>
> As for whether any of our results cast doubt on the conjecture: it is worth noting that for general compression schemes, we have a compression for multiclass classification (with finite graph dimension) that depends on the number of labels. If this dependence can be eliminated in the general case, the multiclass and binary settings would exhibit similar behavior.

---

### Official Review · Reviewer_wMsz · 2024-11-11
**Review of "Sample Compression Scheme Reductions"**

**Rating:** 7
**Confidence:** 4

**Review:**

This paper develops sample compression algorithms for a handful of learning settings via explicit reductions to the simpler (and better understood) binary classification setting. The authors investigate 3 learning settings: 1) Multiclass classification, 2) bounded regression with lp losses and 3) adversarial robust learning. Their main contribution is to show that in these settings there is a simple, yet general, argument of the following form: for given (multiclass, regression etc) function class $\mathcal{F}$ with complexity parameter $d$, there is an associated binary function class $\mathcal{F}'$ with vc dimension $d$ such that a compression scheme for $\mathcal{F}'$ with compression size $f(d)$ can be transformed into a compression scheme for the original non-binary class F with compression size $O(f(d)\log(|\mathcal{Y}|)$ where $|\mathcal{Y}|$ is the number of labels (or a comparable notion for regression and adversarial robust learning). Furthermore, given more ``structured'' guarantees on the binary compression scheme e.g. 1) the reconstructed function corresponds to a function in the class or a majority vote of functions in the class or 2) the sample compression scheme is \emph{stable}, they are able to show that this addition $O(\log(|\mathcal{Y}|)$ factor can be removed providing an even sharper bounds for the original function class.

The authors also makes other smaller contributions such as: 1) introducing an ``infinitized'' notion of sample compression for infinite sequences and showing a simple constructions based on Littlestone's Standard Optimal Algorithm and 2) constructing an adversarially robust learnable class that has no finite compression scheme by porting a construction for a similar impossibility in the partical concept class setting to theirs.

I think this a good paper that is well written and easy to follow. The authors do a great job in explaining how their results compare to existing results in the literature. I consider that the most general reduction technique that the authors propose with no assumption on the relevant binary compression schemes to be folklore. However, the inclusion of this result definitely aids in understanding (and appreciating) how the more delicate reduction works when there is a structural assumption on the compression scheme (stable compression scheme or majority of functions in the base class). I believe the reductions with more structural assumptions are much more interesting/novel as they correspond to some of the most well known compression schemes as the authors mention. I also found the results around the infinitized compression schemes as well as the impossibility result for adversarial robust compression scheme to be very standard.


Some minor comments on presentation:
In the proof of theorem 7, in the last sentence of the first paragraph (right detalining the compression and reconstruction steps) the authors write Thus $c(x,y) =1$ where I believe they meant $g(x,y) =1$. Actually, I find the notation that uses g for the reconstructed function to be confusing as it clashes with the notation g_c for functions in the ``binarized'' class $\mathcal{C}_{\mathcal{Y}}$.

In the first display of the proof of theorem 9 the authors define a set $\kappa_b'(\mathcal{S}_{\mathcal{Y}})$ but the definition is extrmeely hard to parse (and there is a typo with the last closing bracket being "]" rather than ")" ).

The proofs of the reductions in section 5 do not match the style of the proofs in the other sections e.g. the compression and reconstruction steps are not spelled out like in the previous sections.

**Paper Award:**

No

---

> ### Author Response · Authors · 2024-11-25
> **Thank you for the review.**
>
> We thank the reviewer for the positive and thoughtful feedback and insightful suggestions.
>
> Minor comments about notations and typos in the proofs of Theorems 7 and 9, and Section 5:
> We thank the reviewer for their careful reading and helpful suggestions! We agree with the comments and will make the changes accordingly.

---

### Official Review · Reviewer_ysvY · 2024-11-17
**Accept -- Interesting collection of results**

**Rating:** 8
**Confidence:** 4

**Review:**

The paper develops sample compression schemes for multiclass classification, regression, and adversarially robust learning. In particular, the paper provides novel reductions from sample compression schemes from these settings to binary classification schemes. It builds on the longstanding open problem in machine learning related to the sample compression conjecture, which proposes that any binary concept class with a finite VC dimension should have a compression scheme of size O(d_{VC}). At a high level, the authors show that :

- if binary classification has a sample compression scheme of size f(d_VC), then multiclass classification has a compression scheme of size O(f(d_G \log(|Y|))) where Y is the set of the label and   d_G  is the graph dimension of the considered class.

- Under further assumptions, such as the reconstruction function (in the sample compression scheme for binary classification) being stable, based on the majority vote of classifiers, or proper, this extra log(|Y|) factor can be avoided. This circumvents a recently established lower bound due to Pabbaraju et al. 2024.

-Similar reductions and separation are also provided for regression, and adversarially robust learning settings.

- The authors introduced infinitized and inflated compression schemes to handle infinite data sequences, which I found to be novel; I was not aware of prior works discussing this.

- The authors leave interesting open problems about whether compression schemes for regression can be improved using fat-shattering dimensions, instead of the pseudo dimension (currently considered in the paper).

Overall the proofs seem to be straightforward. I see this paper as a collection of various small results on the same theme. Taken together they are sufficient for a good ALT paper. I recommend acceptance. This paper will serve as a useful tool when we finally resolve the VC sample compression conjecture.

Questions / Cons:
Can the authors discuss why is it even interesting to consider infinitized setting? It seems like an artificial construction.
Can authors provide a discussion on the tightness of the converse direction, i.e. binary classification setting to the setting considered in the paper (multiclass, regression, adversarially robust learning, etc). Can the two directions be used to connect the seemingly unrelated problems of multiclass classification or regression to each other in a tight manner?

**Paper Award:**

No

---

> ### Author Response · Authors · 2024-11-25
> **Thank you for the review.**
>
> We thank the reviewer for their positive and thoughtful feedback, as well as the insightful suggestions.
>
> Question: Can the authors discuss why it is even interesting to consider an infinitized setting?
>
> Answer: We introduced this concept to address the general multiclass setting and regression. We believe it represents an interesting mathematical object in its own right, especially since natural classes (with finite Littlestone dimension) can admit such a compression. Furthermore, it exhibits a different structure compared to finite compression schemes.
>
> Question: Can the authors provide a discussion on the tightness of the converse direction, i.e., binary classification setting to the setting considered in the paper (multiclass, regression, adversarially robust learning, etc.)?
>
> Answer: The reductions in this direction are relatively straightforward, as binary classification is a special case of multiclass classification, regression with the  $\ell_1$ loss, and adversarial learning when  $U(x) = x.$
> (We hope we have interpreted your question correctly.)
>
> Question: Can the two directions be used to connect the seemingly unrelated problems of multiclass classification or regression to each other in a tight manner?
>
> Answer: This is a good point. We can reduce the regression setting (assuming finite pseudo-dimension) to multiclass classification (assuming finite graph dimension). However, the reverse direction is more challenging. In the standard multiclass setting, we use the crude zero-one loss, whereas, in regression with the  $\ell_p$ loss, the distance to the true label plays a significant role.

---

### Meta-Review · Area_Chair_X2bi · 2024-12-14

**Recommendation:** Accept
**Confidence:** 4

**Metareview:**

The paper offers tools for contructing compression schemes for various learning problems beyond binary classification,
by reducing these tasks to compression schemes for binary classification. The paper is well written and sems to be mathematically solid
and the reviewrs are enthusiastic about it. The min weakness of this submission is the problem discusses is of rather secondary significance,
at least as long as the major binary classification sample compression conjecture remains unproved.

**Paper Award:**

No